# Contrastive dimension reduction: when and how?

**Sam Hawke**
Department of Biostatistics
University of North Carolina at Chapel Hill
shawke@unc.edu

**YueEn Ma**
Department of Statistics & Operations Research
University of North Carolina at Chapel Hill
myueen@unc.edu

**Didong Li**
Department of Biostatistics
University of North Carolina at Chapel Hill
didongli@unc.edu

## Abstract

Dimension reduction (DR) is an important and widely studied technique in exploratory data analysis. However, traditional DR methods are not applicable to datasets with a contrastive structure, where data are split into a foreground group of interest (case or treatment group), and a background group (control group). This type of data, common in biomedical studies, necessitates contrastive dimension reduction (CDR) methods to effectively capture information unique to or enriched in the foreground group relative to the background group. Despite the development of various CDR methods, two critical questions remain underexplored: when should these methods be applied, and how can the information unique to the foreground group be quantified? In this work, we address these gaps by proposing a hypothesis test to determine the existence of contrastive information, and introducing a contrastive dimension estimator (CDE) to quantify the unique components in the foreground group. We provide theoretical support for our methods and validate their effectiveness through extensive simulated, semi-simulated, and real experiments involving images, gene expressions, protein expressions, and medical sensors, demonstrating their ability to identify the unique information in the foreground group.[1]

## 1 Introduction

High-dimensional datasets are ubiquitous in the era of big data, arising from applications such as image data (Shorten and Khoshgoftaar, 2019), gene/protein expressions (Bhola and Singh, 2018), and medical data from wearable devices (Cho et al., 2021; Banaee et al., 2013), to name a few. Dimension reduction (DR) is crucial in these contexts for various reasons (Fan et al., 2014): improving data visualization, reducing computational cost for downstream analysis (Fan and Li, 2006), minimizing noise (Thudumu et al., 2020), and enhancing interpretability (Johnstone and Titterington, 2009).

DR has been a vibrant research area for several decades, leading to the development of numerous methods (Ayesha et al., 2020). Principal component analysis (PCA, Hotelling (1933)), for instance, maximizes the variance in the reduced space and is widely applied across different domains (Jirsa et al., 1994; Novembre and Stephens, 2008; Pasini, 2017). Other methods, such as Multi-Dimensional Scaling (MDS, Torgerson (1952)) and Isomap (Tenenbaum et al., 2000), focus on preserving pairwise Euclidean and geodesic distances between samples, respectively. Probabilistic PCA (PPCA, Tipping and Bishop (1999)) incorporates uncertainty quantification through a probablistic model. Additionally,

---

[1]All code available at https://github.com/myueen/contrastive-dimension-estimation

38th Conference on Neural Information Processing Systems (NeurIPS 2024).

techniques such as t-distributed Stochastic Neighborhood Embedding (tSNE, Van der Maaten and Hinton (2008)) and Uniform Manifold Approximation (UMAP, McInnes et al. (2018)), preserve the local probability distribution and manifold topological structure in the data, respectively.

In recent years, a new type of data has garnered attention, particularly in biomedical research involving case-control studies. Here, data are divided into a foreground group (case or treatment group) and a background group (control group). The objective is to identify low-dimensional representations unique to or enriched in the foreground group. This scenario necessitates contrastive dimension reduction (CDR) methods. For example, in a benchmark dataset for CDR on mouse protein expressions (Higuera et al., 2015), the foreground group consists of mice who received a shock therapy, while the background group includes mice that did not. The goal is to isolate the protein expression patterns unique to the foreground group, i.e., the shocked mice.

To address this problem, several CDR methods have been developed. For example, Contrastive PCA (CPCA, Abid et al. (2018)) extends PCA by maximizing the variation in the foreground while minimizing variation in the background. Additionally, Probabilistic Contrastive PCA (PCPCA, Li et al. (2020)) generalizes both CPCA and PPCA by incorporating a probabilistic model into the contrastive setting, enabling statistical inference. Furthermore, the Contrastive Latent Variable Model (CLVM, Severson et al. (2019)) introduces latent variables specifically designed for this contrastive setting. Moreover, the Contrastive Variational Autoencoder (CVAE, Abid and Zou (2019)) and Contrastive Variational Inference (CVI, Weinberger et al. (2023)) both leverage deep learning approaches to detect nonlinear patterns in the data. Collectively, these methods aim to isolate patterns unique to the foreground data relative to the background. These methods have been successful with a variety of datasets, identifying patterns that are often scientifically meaningful and opening doors to new scientific discoveries. For example, studies using these methods on mouse protein expression data have successfully identified subgroup structures, such as distinguishing between mice with and without Down Syndrome, and highlighted key proteins responsible for these differences. We reference various applications and discoveries using these methods in section 6 and appendix D.

Despite the development of these CDR methods, two critical questions remain underexplored: when should these CDR methods be applied, and how many unique components exist in the foreground group? The first question can be reframed as follows: given foreground and background groups, can we determine whether there exists unique information in the foreground group? The second question pertains to estimating the number of unique representations to the foreground, termed contrastive dimension in this paper. This is a counterpart of the intrinsic dimension estimation problem, with a rich literature (Camastra and Staiano, 2016; Levina and Bickel, 2004), but in the contrastive context.

In this paper, we answer the first question by proposing a hypothesis test to determine the existence of unique information in the foreground data. To answer the second question, we introduce the rigorous notion of contrastive dimension and provide a consistent estimator of this quantity. We provide theoretical support of our methods and validate their effectiveness through extensive simulated, semi-simulated, and real experiments involving images, gene expressions, protein expressions, and medical sensors.

The rest of the paper is organized as follows. In section 2, we provide more background on CDR methods and present our definition of contrastive dimension. In section 3, we present our solutions to the above questions: a hypothesis test and an estimator for the contrastive dimension. In section 4, we prove the consistency of our proposed estimator of the contrastive dimension and establish a finite-sample error bound. We demonstrate the effectiveness of the methods on simulated and semi-simulated data in section 5 and real data in section 6. Finally, in section 7, we discuss the strengths and weaknesses of our methods and identify potential directions for further study.

## 2   Background

In this section, we introduce our notations and provide a more detailed overview of various existing CDR methods. Throughout this paper, we denote the foreground data as $X = \{x_1, \cdots, x_{n_x}\} \subset \mathbb{R}^p$ with dimension $p$ and sample size $n_x$, and background data as $Y = \{y_1, \cdots, y_{n_y}\} \subset \mathbb{R}^p$ with the same dimension $p$ but potentially different sample size $n_y$. We do not make any assumptions on the ratio of sample sizes between groups, $n_x : n_y$. For simplicity, we also assume each dataset is centered separately. Next, we briefly overview some existing CDR methods.

**Contrastive Principal Component Analysis (CPCA).** CPCA (Abid et al., 2018) was proposed to uncover the low-dimensional structure that is unique to the foreground data $X$ relative to the

background data $Y$. Taking one-dimensional CPCA as an example, the objective is to find a direction, represented by a unit vector $v \in \mathbb{R}^p$, that explains more variance in the foreground among all directions, and less variance in the background. Let $C_X = \frac{1}{n_x} \sum_{i=1}^{n_x} x_i x_i^\top$ and $C_Y = \frac{1}{n_y} \sum_{j=1}^{n_y} y_j y_j^\top$ be the sample covariance matrices of two datasets. Then, CPCA solves the following optimization problem:

$$\underset{\|v\|=1}{\operatorname{argmax}} \, v^\top C_X v - \gamma v^\top C_Y v = \underset{\|v\|=1}{\operatorname{argmax}} \, v^\top C v,$$

where $\gamma \in [0, \infty]$ is a tuning parameter, called the contrastive parameter, and $C = C_X - \gamma C_Y$ is called the contrastive covariance matrix. The optimal $v$ is simply the top eigenvector of $C$. When $\gamma = 0$, CPCA coincides with PCA, and when $\gamma = \infty$, CPCA finds the direction that explains the least variance in the background group.

**Contrastive Latent Variable Model (CLVM).** CLVM (Severson et al., 2019) is a latent variable model, similarly proposed to discover patterns enriched in the foreground group $X$:

$$x_i = Sz_i + Wt_i + \varepsilon_i, \; y_j = Sz_j + \varepsilon_j,$$

where $z_i, z_j \in \mathbb{R}^k$ and $t_i \in \mathbb{R}^t$ are the latent variables, and $\varepsilon_i, \varepsilon_j \in \mathbb{R}^p$ are the noise terms. The factor loading $S \in \mathbb{R}^{p \times k}$ represents the space shared between the foreground and background groups, while the factor loading $W \in \mathbb{R}^{p \times t}$ represents the space unique to the foreground group. The primary goal of CLVM is to identify W, which represents the foreground-specific information.

**Probabilistic Contrastive Principal Component Analysis (PCPCA).** PCPCA (Li et al., 2020) was proposed as a probabilistic model to extend CPCA, allowing for uncertainty quantification:

$$x = Wz_x + \varepsilon_x, y = Wz_y + \varepsilon_y,$$

where $W$ represents the factor loading, and $\varepsilon_x, \varepsilon_y$ are noise terms. However, PCPCA utilizes an unconventional strategy for estimating $W$. Instead of maximizing the joint likelihood, which would be appropriate if the foreground and background groups are assumed to share the same space, the PCPCA objective is to maximize $\frac{p(X|W)}{p(Y|W)^\gamma}$, where $\gamma \in [0, \infty]$ is the contrastive hyperparameter, as in CPCA. When $\gamma = 0$, PCPCA reduces to Probablistic PCA (Tipping and Bishop, 1999) on the foreground data, while PCPCA reduces to CPCA as the noise level goes to zero.

**Contrastive VAE (CVAE).** CVAE (Abid and Zou, 2019) considers a broadened view of the data-generating process, in which the foreground data $X$ follows an arbitrary probability distribution $f$ with unknown parameters $\theta$, conditional on salient variables $s$ and irrelevant variables $z$. The background data $Y$ are assumed not to have the salient variables, as the CVAE model is given by

$$x_i \sim f_\theta(\cdot | s_i, z_i), \; y_j \sim f_\theta(\cdot | 0, z_j').$$

The problem here is to learn the parameters $\theta$ of $f$, which CVAE does by training two probabilistic encoders $q_{\phi_s}(s|x)$ and $q_{\phi_z}(z|x)$ (salient and irrelevant) to infer $s$ and $z$, respectively, from the observed features, and a decoder network $f_\theta(\cdot)$ which reconstructs the original samples from $x = [s, z]$ in the foreground data, or $y = [0, z']$ in the background data.

**Contrastive Variational Inference (CVI).** CVI was proposed in Weinberger et al. (2023) specifically for the purpose of disentangling the split between treatment and control groups for single-cell data. To this end, CVI posits a data-generating process with latent variables $z_n$, representing shared information between the two groups, and $t_n$, representing information unique to the foreground group. Each gene expression in each sample of the foreground group is assumed to follow a generative process depending on $z_n$ and $t_n$ (both with $\mathcal{N}(0, I)$ priors), but in the background group $t_n = 0$ is assumed, similarly to CVAE. The posterior distribution of CVI is analytically intractable, so it is approximated using variational inference.

Each of the methods above has a similar goal: to discover variation unique to the foreground group relative to the background. However, none of these methods enables detection of whether there exists such variation at all. Additionally, for each method, the assumed reduced dimension (or number of latent features unique to the foreground) was treated as a tuning parameter required as input to run the method, without an easy-to-implement method to choose such a dimension.

Motivated by this gap, we consider the following linear model:

$$x_i = z_i + \varepsilon_i, \; y_j = w_j + \varepsilon_j, \; i = 1, \ldots, n_x, \; j = 1, \ldots, n_y \tag{1}$$

where $z_i \in V_x$, $w_j \in V_y$ for some low-dimensional linear subspaces $V_x, V_y \subset \mathbb{R}^p$, and $\varepsilon_i, \varepsilon_j$ represent the noise. We let $x$ represent foreground and $y$ represent background.

However, defining and identifying the information that is truly unique to the foreground requires a more rigorous approach. For example, in CLVM, even if we estimate a nonzero $W$, it may not present information unique to the foreground if $S$ and $W$ are not appropriately distinguished. This motivates the need for a precise definition of the contrastive subspace and contrastive dimension:

**Definition 1.** *Under the above notation, we define the contrastive subspace to be* $V_{xy} := \mathrm{Proj}_{V_y^\perp}(V_x)$ *and the contrastive dimension, denoted by* $d_{xy}$*, to be the dimension of the contrastive subspace:* $d_{xy} := \dim(V_{xy})$*.*

The rationale of our definition is that the contrastive subspace contains the information unique to the foreground data, guaranteed by the projection to $V_y^\perp$, the orthogonal complement of $V_y$. In order to determine the existence of unique information in the foreground group, it is desirable to have a hypothesis test for testing $H_0 : d_{xy} = 0$ versus $H_1 : d_{xy} > 0$. Furthermore, an estimator for $d_{xy}$ is valuable for providing a suggestion to use for the reduced dimension parameter in the CDR models presented above.

**Remark 1.** *Based on model 1 and definition 1, $V_x \subset V_y$ is equivalent to $d_{xy} = 0$.*

**Remark 2.** *Even a minute departure of $V_x$ from $V_y$ results in a nonzero $d_{xy}$.*

Because we are interested in detecting where the foreground differs from the background, we allow our framework to detect even a small departure of $V_x$ from $V_y$, although a bigger departure will be easier to detect. This motivates our study of the principal angle between $V_x$ and $V_y$ in section 3.

It is worth noting that the new tasks we are proposing, namely determining whether there is unique information in the foreground group and quantifying how much, depart from the traditional CDR task of estimating the space representative of such unique information, assuming it exists. In this paper, we focus on our solution to these new tasks, starting with manageable assumptions about the data.

# 3 Methods

In this section, we present our methods to address the questions of whether there exists unique information in the foreground group and how many dimensions there are unique to the foreground group. First, we detail the hypothesis test of $H_0 : d_{xy} = 0$ versus $H_1 : d_{xy} > 0$. We introduce a contrastive version of the bootstrap resampling scheme, based on the null assumption that $d_{xy} = 0$, which is equivalent to the assumption of $V_x \subset V_y$. Next, we introduce the CDE of $d_{xy}$. In both methods, we make the very reasonable sample size assumptions that $n_x > d_x$ and $n_y > d_y$, as many existing studies have observed a small intrinsic dimension (Pope et al., 2021). The inference in both of these problems is based on the notion of principal angles between two linear subspaces (Ye and Lim, 2016):

**Definition 2.** *Let $U$ and $V$ be two subspaces of $\mathbb{R}^p$ with dimensions $d_1$ and $d_2$, then the principal angles, denoted by $\theta_i(U, V)$, $i = 1, \cdots, \min(d_1, d_2)$ are defined recursively as follows*

$$\cos(\theta_i(U, V)) = \max_{u_i \in U, v_i \in V, \|u_i\| = \|v_i\| = 1, u_i^\top u_j = v_i^\top v_j = 0, \forall j < i} u_i^\top v_i$$

To be clear, the $u_i$ and $v_i$ solving the maximization problem are not uniquely defined; however, the principal angles $\theta_i$ are uniquely defined. The principal angles measure the "difference" between two subspaces. For instance, if two spaces coincide, all principal angles are zero; when two spaces are orthogonal to each other, all principal angles are $\pi/2$. If $U_0$ and $V_0$ are orthogonal matrices spanning $U$ and $V$, respectively, then the calculation of the principal angles is given by the following equation

$$\cos(\theta_i(U, V)) = \sigma_i(U_0^\top V_0), \tag{2}$$

where $\sigma_i(\cdot)$ denotes the $i$-th singular value of a matrix. As a result, we adopt this notion to study the relation between $V_x$ and $V_y$ for our purposes.

## 3.1 Hypothesis test

We develop a hypothesis test for testing $H_0 : d_{xy} = 0$ versus $H_1 : d_{xy} > 0$ under the version of model 1 with the following assumptions,

$$x_i = S_x z_i + \varepsilon_i, \ y_j = S_y w_j + \varepsilon_j, \tag{3}$$

where $S_x \in \mathbb{R}^{p \times d_x}$ and $S_y \in \mathbb{R}^{p \times d_y}$ are full rank, $z_i \sim \mathcal{N}_{d_x}(0, I)$, $w_i \sim \mathcal{N}_{d_y}(0, I)$, $\mathbb{E}[\varepsilon_i] = \mathbb{E}[\varepsilon_j] = 0$, $\mathrm{Cov}(\varepsilon_i) = \sigma_x^2 I$, and $\mathrm{Cov}(\varepsilon_j) = \sigma_y^2 I$.

First, we assume $\max(d_x, d_y) < p/2$, which is a very weak assumption given that $p$ is often much bigger than the intrinsic dimension in high-dimensional datasets.

**Remark 3.** *With model 1, we do not make any assumptions the relations between intrinsic dimensions $d_x$ and $d_y$. However, by definition 1, $d_x > d_y$ implies that $d_{xy} > 0$, indicating that there must be some unique information in the foreground space. Therefore, for our hypothesis test, we focus on the more challenging but important case of $d_x \le d_y$.*

An equivalent formulation of the null and alternative hypotheses in this model is $H_0 : \mathcal{C}(S_x) \subset \mathcal{C}(S_y)$ versus $H_1 : \mathcal{C}(S_x) \not\subset \mathcal{C}(S_y)$, where $\mathcal{C}(\cdot)$ denotes the column space of a matrix. Motivated by the assumption that $V_x \subset V_y$ under $H_0$, we draw a sample of size $n_x$ with replacement from $\{x_1, \ldots, x_{n_x}\}$, and we draw a sample of size $n_y$ with replacement from $\{x_1, \ldots, x_{n_x}, y_1, \ldots, y_{n_y}\}$. We repeat this bootstrap procedure $B$ ($B = 1000$ throughout this paper) times and compare the largest angle between $\widehat{V}_x$ and $\widehat{V}_y$ in the original data with the distribution of the largest angles between those subspaces in the resampled data. The exact calculation is deferred to section 3.2.

---

**Algorithm 1:** Contrastive Bootstrap Hypothesis Test

---

**Input:** Foreground $\{x_i\}_{i=1}^{n_x}$, Background $\{y_j\}_{j=1}^{n_y}$, dimensions $d_x \le d_y$, bootstrap parameter $B$
**Output:** p-value for test of $H_0 : V_x \subset V_y$ in model 1

1   $\widehat{\lambda}_1 \leftarrow \widehat{\lambda}_1(x, y, d_x, d_y)$ ;          `// Compute as in Algorithm 2`
2   **for** $b = 1$ **to** $B$ **do**
3      $x_* \leftarrow$ sample with repl.$(x, n_x)$ ;          `// Resample foreground`
4      $y_* \leftarrow$ sample with repl.$(\{x, y\}, n_y)$ ;     `// Resample background, pooled`
5      $\lambda_b^* \leftarrow \widehat{\lambda}_1(x_*, y_*, d_x, d_y)$ ;          `// Compute as in Algorithm 2`
6   $p \leftarrow \frac{\#\{\lambda_b^* < \widehat{\lambda}_1\}}{B}$
7   **return** $p$

---

We demonstrate with simulations in section 5 that this hypothesis test produces conservative results, and in section 7 we discuss a potentially more powerful likelihood-based alternative. Additionally, this hypothesis test requires as input intrinsic dimension estimates for $d_x$ and $d_y$, for which there exists a rich literature with a variety of sophisticated methods (Campadelli et al., 2015). This conservatism is designed to reduce the likelihood of false positives, but it can also lead to some signals not being detected by the hypothesis test. One explanation for this phenomenon could be that the test's resampling technique might not handle nonlinear structures in the data effectively (e.g. if the data lie on a nonlinear manifold). Because $V_x$ and $V_y$ are assumed to be linear subspaces, a nonlinear pattern can obscure the differences the test is designed to detect.

## 3.2 Contrastive dimension estimator (CDE)

We start from the following lemma to link our target, $d_{xy}$, and principal angles between $V_x$ and $V_y$.
**Lemma 1.** $d_{xy} = \dim(\mathrm{Proj}_{V_y^\perp} V_x) = \#\{i : \theta_i(V_x, V_y) > 0\} + \max(d_x - d_y, 0)$.

As a result, to construct an estimator of $d_{xy}$, it suffices to estimate $V_x$, $V_y$, and calculate the corresponding principal angles. Based on eq. (1), $V_x$ and $V_y$ consist of the top $d_x$ and $d_y$ eigenvectors of $\Sigma_x$ and $\Sigma_y$, respectively, which can be estimated by sample covariance matrices $\widehat{\Sigma}_x := \frac{1}{n_x} \sum_{i=1}^{n_x} x_i x_i^\top$ and $\widehat{\Sigma}_y := \frac{1}{n_y} \sum_{j=1}^{n_y} y_j y_j^\top$. Thus, we propose the following estimator:

$$\widehat{d}_{xy} := \# \left\{ i : \theta_i \left( \mathrm{eig}_{1:d_x} \left( \widehat{\Sigma}_x \right), \mathrm{eig}_{1:d_y} \left( \widehat{\Sigma}_y \right) \right) > \epsilon \right\} + \max(d_x - d_y, 0), \tag{4}$$

where $\epsilon > 0$ is a small toleration due to randomness from finite samples, and $\text{eig}_{1:d}$ refers to the span of the first $d$ eigenvectors. Let $\widehat{U}_x := \text{eig}_{1:d_x}\left(\widehat{\Sigma}_x\right)$ and $\widehat{U}_y := \text{eig}_{1:d_y}\left(\widehat{\Sigma}_y\right)$, by eq. (2), we have $\lambda_i := \cos(\theta_i) = \sigma_i(\widehat{U}_x^\top \widehat{U}_y)$, leading to algorithm 2.

---

**Algorithm 2:** Contrastive Dimension Estimator

**Input:** Foreground $\{x_1, \ldots, x_{n_x}\}$, Background $\{y_1, \ldots, y_{n_y}\}$, dimensions $d_x, d_y$, tolerance $\epsilon$

**Output:** $\widehat{d}_{xy}$

1  $(\widehat{\Sigma}_x, \widehat{\Sigma}_y) \leftarrow (n_x^{-1} \sum_{i=1}^{n_x} x_i x_i^\top, \ n_y^{-1} \sum_{i=1}^{n_y} y_i y_i^\top)$ ;      `// Compute sample covariances`

2  $(\widehat{U}_x, \widehat{U}_y) \leftarrow \left(\text{eig}_{1:d_x}(\widehat{\Sigma}_x), \text{eig}_{1:d_y}(\widehat{\Sigma}_y)\right)$ ;      `// Compute sample eigenvectors`

3  $U D V^\top \leftarrow \widehat{U}_x^\top \widehat{U}_y$ ;      `// SVD`

4  $(\widehat{\lambda}_1, \ldots, \widehat{\lambda}_{\min\{d_x, d_y\}}) \leftarrow \text{diag}(D)$ ;      `// Singular values`

5  $\widehat{d}_{xy} \leftarrow \#\{j : \widehat{\lambda}_j < 1 - \epsilon\} + \max(d_x - d_y, 0)$

6  **return** $\widehat{d}_{xy}$

---

Note that algorithm 2 takes intrinsic dimensions $d_x$ and $d_y$ as input. In practice, the user must choose a method to estimate the intrinsic dimension of both the foreground and background datasets. This flexibility allows for the incorporation of different intrinsic dimension estimators to complement our approach (Bac et al., 2021; Pope et al., 2021).

Some users may only be interested in estimating the contrastive dimension $\widehat{d}_{xy}$ and not in the hypothesis test for $d_{xy} = 0$. These users can skip the hypothesis testing step. However, the hypothesis test offers a direct way to assess uncertainty, guiding users on whether it is worthwhile to proceed with contrastive dimension reduction methods.

In the next section, we study the asymptotic and finite-sample behavior of this estimator $\widehat{d}_{xy}$.

## 4   Consistency and finite sample error bound

In this section, we provide theoretical support for the estimator $\widehat{d}_{xy}$ presented above, first by establishing its consistency, and then by constructing a finite-sample error bound for it.

**Theorem 1.** *Assume that the second moments $\Sigma_x$ and $\Sigma_y$ are finite for both groups, and that the top $d_x + 1$ eigenvalues of $\Sigma_x$ and top $d_y + 1$ eigenvalues of $\Sigma_y$ are distinct. Then, our proposed estimator $\widehat{d}_{xy}$ is consistent: $\widehat{d}_{xy} \xrightarrow[n_y \to \infty]{n_x \to \infty} d_{xy}$.*

The assumption of distinct eigenvalues is common in the literature (Vershynin, 2018) and ensures the identifiability of eigenvectors for each covariance matrix. Because we are interested in convergence to the space spanned by the top eigenvectors, rather than individual eigenvectors themselves, this assumption might be relaxed to the eigengap assumption that the $d_x$th and $(d_x + 1)$th eigenvalues of $\Sigma_x$ are distinct, and that the $d_y$th and $(d_y + 1)$th eigenvalues of $\Sigma_y$ are distinct. See section 7 for more discussion.

To describe the finite sample error bound, we recall that $U_x$ and $U_y$ are spanned by the top $d_x$ and $d_y$ eigenvectors of $\Sigma_x$ and $\Sigma_y$, respectively, and singular values of $W := U_x^\top U_y$, denoted by $\lambda_i = \sigma_i(W), i = 1, \cdots, m$, where $m = \min(d_x, d_y)$, are the cosines of principal angles between $U_x$ and $U_y$. In practice, estimating the contrastive dimension as an integer can be somewhat restrictive for providing a finite sample error bound. Instead, we report the singular values to account for more uncertainty and provide a more nuanced understanding. Thus, we present the following finite sample error bound (and hence the convergence rate) of these estimated singular values rather than $\widehat{d}_{xy}$.

**Theorem 2.** *In addition to all assumptions in theorem 1, we assume $x$ and $y$ in model 1 are sub-Gaussian. More precisely, assume that there exist constants $K_x, K_y \geq 1$ such that*

$$\|\langle x, v\rangle\|_{\psi_2} \leq K_x \mathbb{E}\left[\langle x, v\rangle^2\right] \text{ and } \|\langle y, v\rangle\|_{\psi_2} \leq K_y \mathbb{E}\left[\langle y, v\rangle^2\right] \text{ for any } v \in \mathbb{R}^p$$

where $\|\cdot\|_{\psi_2}$ is defined as in Section 3.4 of Vershynin (2018). Then, for any $u > 0$, with probability at least $1 - 2e^{-u}$, we have

$$\max_{j=1,\ldots,m} \left|\widehat{\lambda}_j - \lambda_j\right| \le C \sum_{k\in\{x,y\}} \sqrt{d_k}\delta_k^{-1}K_k^2 \left(\sqrt{\frac{p+u}{n_k}} + \frac{p+u}{n_k}\right) \|\Sigma_k\| = O(n_x^{-1/2} + n_y^{-1/2}) \tag{5}$$

where $C$ is an absolute, positive constant, $\delta_\bullet$ is the minimum eigengap among the top $d_\bullet + 1$ eigenvalues of $\Sigma_\bullet$, and $\|\bullet\|$ denotes spectral norm.

To show the finite sample error bound, we need the sub-Gaussianity assumption to ensure that the tails of the distribution are not too heavy. This assumption is also common and encompasses a large class of distributions (Vershynin, 2018).

Having established the theoretical support for the estimator, in the next section we focus on applying our methods to simulated data.

# 5 Simulations

To demonstrate the validity of our methods in practice, we apply them to four simulations for different purposes. The first two examples are purely simulated, fully under our control, with $d_{xy} = 0$ for the first one and $d_{xy} > 0$ for the second one. Because these first two simulations are purely synthetic, we repeat them both 100 times and report the means and standard deviations of the $p$-value, $\widehat{d}_{xy}$, and the four smallest singular values. The third simulation steps away from the ground truth being completely known by combining synthetic image data with known dimension, with real, grassy noise image data, elucidating the methods' performance in an interpretable setting. The final simulation is a variation on the third one, in which the synthetic data are replaced with MNIST images (Deng, 2012).

We describe the experimental settings of the simulations in more detail below and present results, including the $p$-value from our test $\widehat{d}_{xy}$, and the smallest four singular values, as shown in Table 4. We denote the estimates with a hat, if needed. In all these examples, the threshold is $\epsilon = 0.1$.

We do not compare $\widehat{d}_{xy}$ with other estimators of contrastive dimension because, to our knowledge, no such estimators have been proposed in the literature.

Table 1: Summary of simulation results

| Setup | $d_x$ | $d_y$ | $d_{xy}$ | p-value | $\widehat{d}_{xy}$ | 4 smallest singular values |
|---|---|---|---|---|---|---|
| Simulation 1 | 6 | 8 | 0 | 0.814 | 0.44 | 0.901, 0.939, 0.961, 0.975 |
| Simulation 2 | 6 | 8 | 6 | 0.029 | 6 | **0.059**, **0.106**, **0.159**, **0.215** |
| Sim. 3: Disk + Line | 4 | $\widehat{20}$ | 1 | 0.000 | 1 | **0.490**, 0.975, 0.981, 0.988 |
| Sim. 4: MNIST + Line | $\widehat{11}$ | $\widehat{14}$ | 1 | 0.000 | 1 | **0.407**, 0.904, 0.955, 0.975 |

**Simulation 1.** In this first simulation, we generate data from the specific case of model (3) with variance $\sigma_x^2 = \sigma_y^2 = 0.25$, $\mathcal{C}(S_x) \subset \mathcal{C}(S_y)$ such that $d_{xy} = 0$, $n_x = n_y = 100$. Because $d_{xy} = 0$ is the assumption of $H_0$ for the hypothesis test in Section 3, we repeat this simulation 100 times to empirically demonstrate the conservative type I error rate (defined by the proportion of $p$-values below $\alpha$) versus pre-specified $\alpha \in (0, 1)$. We also repeat this simulation but with $n_x = n_y = 200$, and we present both plots of empirical type I error rate in Figure 1.

As shown in the first row of Table 4, both $\widehat{d}_{xy}$ and the $p$-value agree, on average, with the ground truth that $d_{xy} = 0$, showing that in this simulated setting, our methods do not erroneously detect a signal when there is none. However, given that the average smallest singular value value observed is $0.901$, this simulation suggests that it may be prudent to use a choice of $\epsilon$ greater than $0.1$, i.e. a cutoff for the singular value that is below $0.9$. The standard deviation for $\widehat{d}_{xy}$ among the 100 replications is $0.52$, which also suggests that a lower cutoff may be appropriate. The standard deviation of the $p$-value is $0.153$, and of each of the smallest singular values is $0.025$, $0.013$, $0.009$, and $0.007$, respectively. Both plots of observed type I error in Figure 1 demonstrate that, in this setting, the hypothesis test is conservative. We discuss a more powerful potential alternative in Section 7.

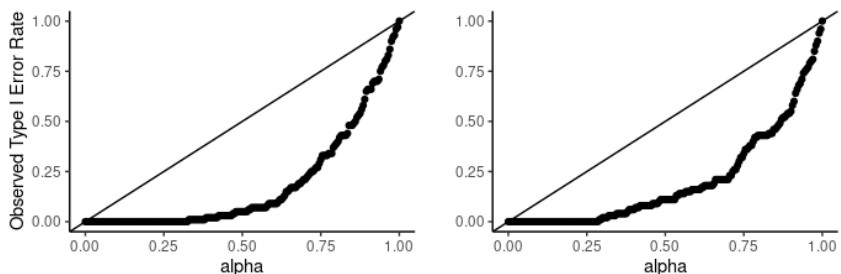

Figure 1: Hypothesis test Type I error plots. Left: $n_x = n_y = 100$. Right: $n_x = n_y = 200$.

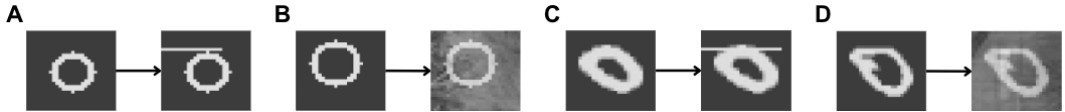

Figure 2: Samples for simulations 3 (AB) and 4 (CD). AC: add line. BD: add grass.

**Simulation 2.** The only difference from Simulation 1 is that $d_{xy} = 6$. Because the singular values were all well below the cutoff of 0.9, the standard deviation of $\hat{d}_{xy}$ among the 100 trials was 0; every trial achieved the correct estimate. Our hypothesis test produced a small $p$-value of 0.029 on average (standard deviation 0.029), which at a significance level of $\alpha = 0.05$ leads one to the correct conclusion of rejecting $H_0$ and concluding that $d_{xy} > 0$. Moreover, the six smallest singular values are $0.059, 0.106, 0.159, 0.215, 0.279, 0.367$ (standard deviations $0.023, 0.028, 0.032, 0.034, 0.038$, and $0.050$, respectively), all well below the threshold of 0.9, illustrating that CDE correctly identifies 6 directions unique to the foreground data.

**Simulation 3.** It is also useful to evaluate the performance of our methods on datasets where the ground truth is only partially known. So, in this simulation, we construct a semi-synthetic image dataset by generating 5000 disks of random centers $(c_x, c_y) \in \{5, \ldots, 24\}^2$ and radii $r \in \{1, \ldots, 10\}$ uniformly, in the 28 by 28 grayscale pixel space.

Of these 5000 images, we draw a white line of random length beginning at the leftmost pixel of the sixth row on 2500 images to create the foreground (Fig. 2A). As a result, $d_x = 4$, parameterized by $c_x, c_y, r$, and length of line. For background, we superimpose the remaining 2500 images on grass images as noise (Fig. 2B) so that $d_y > d_x$. Although $d_y$ is not known, $d_{xy}$ is known to be one (the unique parameter being the length of the line). Here, we used the method of moments estimator (Amsaleg et al., 2018), leading to $\hat{d}_y = 20$.

Our method only gives one singular value (0.529) below the cutoff of 0.9, leading to a correct estimate of $\hat{d}_{xy} = 1$. Additionally, our hypothesis test produces a $p$-value of 0, which suggests to reject $H_0$ in favor of $H_1 : d_{xy} > 0$. This simulation illustrates that our methods produce favorable results when one of the groups has an estimated intrinsic dimension.

**Simulation 4.** To assess our methods in a simulation setting where both $d_x$ and $d_y$ are unknown, we consider a very similar simulation to the previous one, but where we start with 5923 handwritten 0's from the MNIST dataset (Deng, 2012) instead of randomly generated disks. We further randomly split them into two sets, to create foreground by adding lines (Fig. 2C), and background by imposing on grass images (Fig. 2D), leading to the corrupted MNIST data proposed in Abid et al. (2018). Using method of moments, $\hat{d}_x = 11$ and $\hat{d}_y = 14$. Crucially, the contrastive dimension $d_{xy} = 1$ is still known, allowing us to assess the results of our methods when $d_x$ and $d_y$ are both unknown.

CDE again accurately estimates exactly one singular value (0.256) below the cutoff of 0.9, leading to a correct estimate of $\hat{d}_{xy} = 1$. Furthermore, our hypothesis test produces a $p$-value of 0, leading to the correct conclusion that $d_{xy} > 0$ at a significance level of $\alpha = 0.05$. After successful applications of our methods to various settings, it is time to apply them to real datasets.

# 6 Real data experiments

In this section, we apply our methods to real datasets previously studied in the literature to validate their effectiveness and interpret the results. In Table 2, we summarize relevant characteristics of each dataset along with the results. For each dataset, the intrinsic dimension estimation was performed with the method of moments estimator (Bac et al., 2021). Note that in each case where $\widehat{d}_x > \widehat{d}_y$, the $p$-value is not reported because the hypothesis test does not apply. The cutoff for the singular values is set to $0.9$ in all experiments.

Table 2: Summary of Results for Real Data Experiments

| Dataset | p | $n_x$ | $n_y$ | $\widehat{d}_x$ | $\widehat{d}_y$ | p-Value | $\widehat{d}_{xy}$ | 4 Smallest Singular Values |
|---|---|---|---|---|---|---|---|---|
| Noisy MNIST | 784 | 5000 | 5000 | 20 | 20 | 0.008 | 5 | **0.095**, **0.315**, **0.705**, **0.846** |
| Mouse Protein | 77 | 270 | 135 | 5 | 4 | — | 4 | **0.319**, **0.472**, **0.886**, 0.919 |
| mHealth | 23 | 6451 | 3072 | 6 | 10 | 0.000 | 1 | **0.394**, 0.995, 0.997, 0.999 |
| BMMC | 500 | 4501 | 1985 | 26 | 30 | 0.073 | 17 | **0.042**, **0.123**, **0.182**, **0.306** |
| Small Molecule | 1000 | 3096 | 2831 | 12 | 11 | — | 4 | **0.627**, **0.699**, **0.894**, 0.925 |
| ECCITE-Seq | 1000 | 18343 | 2386 | 41 | 32 | — | 13 | **0.850**, **0.857**, **0.874**, **0.877** |
| Pathogen | 1000 | 4481 | 3240 | 24 | 25 | 0.161 | 10 | **0.259**, **0.345**, **0.528**, **0.700** |
| Perturb-Seq | 1000 | 24913 | 8907 | 49 | 47 | — | 28 | **0.038**, **0.102**, **0.131**, **0.176** |
| CelebA: Glasses | 1000 | 13193 | 189406 | 21 | 28 | 0.138 | 1 | **0.215**, 0.972, 0.975, 0.984 |
| CelebA: Hat | 1000 | 9818 | 192781 | 22 | 28 | 0.063 | 2 | **0.702**, **0.803**, 0.924, 0.938 |

**Corrupted MNIST.** The corrupted MNIST dataset was previously studied extensively by Abid et al. (2018), Abid and Zou (2019), Severson et al. (2019), and Li et al. (2020). The foreground group depicts handwritten 0's and 1's corrupted by images of grass, and the background contains only the grassy noise (no handwritten digits). In this case, the digits are unique to the foreground, supported by our $p$-value of $0.008$. Furthermore, the contrastive dimension estimate for this dataset is $\widehat{d}_{xy} = 5$, in line with the commonly believed dimension of digits 0 and 1 (Pope et al., 2021).

**Mouse Protein.** This dataset (Higuera et al., 2015) is a benchmark in the literature, considered in Abid et al. (2018), Li et al. (2020), and Severson et al. (2019). The foreground group contains the protein expressions of mice, some with and some without Down Syndrome, who were subjected to shock therapy, while the background group consists of protein expressions of mice without Down Syndrome who did not receive shock therapy. The intrinsic dimension estimates of $\widehat{d}_x = 5$ and $\widehat{d}_y = 4$ require that $\widehat{d}_{xy} > 0$. In this setting, where the foreground space is bigger than the background space, the foreground group necessarily carries unique information. Nevertheless, the CDE estimate for this dataset of $\widehat{d}_{xy} = 4$, which aligns with the observation of $d_{xy} \geq 2$ in the literature.

**mHealth.** The mHealth dataset (Banos et al., 2015), studied in Abid et al. (2018); Severson et al. (2019), consists of measurements made from medical sensors while a subject performs certain actions. While the foreground group contains measurements taken of a subject either squatting or cycling, the background measures a subject lying down. Our $p$-value $< 0.001$ and $\widehat{d}_{xy} = 1$. This estimate explains the result in Abid et al. (2018), duplicated here in figure 3, showing that a single direction suffices to distinguish subgroups in the foreground data.

**BMMC.** This is a single-cell RNA sequencing dataset (Zheng et al., 2017) studied in Abid et al. (2018), Abid and Zou (2019), Severson et al. (2019), Li et al. (2020), and Weinberger et al. (2023). Here, the foreground group consists of gene expressions measured on bone marrow mononuclear cells (BMMCs) of patients with acute myeloid leukemia before and after receiving a transplant. The background group consists of healthy patients. Previous studies found that the 3rd and 4th PCs can separate pre- and post-transplant groups (Fig. 3), supported by our $\widehat{d}_{xy} = 17 > 4$. However, the $p$-value of $0.073$ stands somewhat in contradiction to this result, maybe due to the conservative nature of this hypothesis test, as illustrated in figure 1 and discussed in section 7.

**Small Molecule.** This is a dataset of cell line responses to small-molecule therapy (McFarland et al., 2020) and used in Weinberger et al. (2023). Here, the foreground group contains measurements from 24 cell lines treated with idasanutlin, while the background are the same cell lines treated with

the control dimethyl sulfoxide. $\widehat{d}_x = 12$ and $\widehat{d}_y = 11$ imply that $\widehat{d}_{xy} > 0$. Nonetheless, $\widehat{d}_{xy} = 4$, suggesting that a CDR method with reduced dimension of 4 is appropriate.

**ECCITE-Seq.** This ECCITE-Seq dataset (Mimitou et al., 2019) was studied in Weinberger et al. (2023), containing joint RNA and surface protein expressions of pooled CRIPSR screens. The foreground group consists of cell transcriptomes of perturbed cells, while the background is a set of control cells. $\widehat{d}_x = 41$ and $\widehat{d}_y = 32$ imply that $\widehat{d}_{xy} > 0$. Therefore, the application of contrastive methods is suitable. Moreover, the CDE estimate of $\widehat{d}_{xy} = 13$ helpfully gauges the number of directions unique to the foreground, recommending a reduced dimension of 13.

**Pathogen Data.** Studied by Weinberger et al. (2023), this dataset (Haber et al., 2017) contains gene expressions in the epithelial cells of mice infected with either Salmonella or H. Poly (foreground) or healthy mice (background). $\widehat{d}_{xy} = 10$, suggesting that a contrastive method might fit well with a reduced dimension of 10. However, the $p$-value of $0.161$ suggests that a linear contrastive method may not be appropriate. In fact, Weinberger et al. (2023) shows that only CVI, a deep learning based approach, works for this dataset, compared with linear methods, which aligns with our results.

**Perturb-Seq:** Studied by Weinberger et al. (2023), this dataset (Adamson et al., 2016) includes gene expressions measured of cells subjected to CRISPR-mediated perturbations (foreground) and cells treated with control guides (background). The hypothesis test is not applicable since $\widehat{d}_x > \widehat{d}_y$. Regardless, CDE provides a helpful estimate for the amount of information unique to the foreground group, suggesting the efficacy of CDR methods.

**CelebA:** The CelebA dataset (Liu et al., 2015) was studied in Abid and Zou (2019) and with images of celebrities. We perform two experiments, with foreground being those wearing glasses and hat, respectively. In both, the background group is the remaining images. In the former case, $\widehat{d}_{xy} = 1$, while in the latter $\widehat{d}_{xy} = 2$. However, both cases have a somewhat large $p$-value, perhaps due to the conservative nature of the hypothesis test or nonlinearity.

## 7    Conclusion and Future Work

In this paper, we have introduced a hypothesis test to determines whether a linear CDR model is appropriate for a given dataset and an estimator of contrastive dimension, representing the dimension of the subspace unique to the foreground group relative to the background. This estimate of contrastive dimension can be used as the reduced dimension in downstream analysis with a CDR method. Meanwhile, there are still some open problems left to future work.

**Intrinsic Dimension Estimator.** The problem of estimating intrinsic dimension is challenging in its own right (Camastra and Staiano, 2016; Campadelli et al., 2015; Fefferman et al., 2016). While methods for it have been developed, there is not a single accepted standard for it. Fortunately, since $\left|\widehat{d}_{xy} - d_{xy}\right| \leq \left|\widehat{d}_x - d_x\right| + \left|\widehat{d}_y - d_y\right|$, our $\widehat{d}_{xy}$ is robust when $d_x$ and $d_y$ are mis-specifying $d_x$ and $d_y$ in the Lipschitz-1 sense.

**Hypothesis Test.** Our hypothesis test introduces the notion of the contrastive bootstrap to allow for resampling under the assumption of the null hypothesis that the foreground space is a subset of the background space. However, a more powerful likelihood ratio test is applicable. Under model 3, the denominator is exactly from the solution of PPCA Tipping and Bishop (1999). However, the numerator involves a nontrivial constrained optimization problem. Naive gradient step method is neither effective nor scalable. An interesting future direction is to resolve this optimization issue.

**Uniqueness of Eigenvalues.** In section 4 we mentioned that the distinctness of eigenvalues ensures the identifiability of eigenvectors for each covariance matrix. However, this assumption can be relaxed since our focus is not on estimating each individual eigenvector but rather on identifying the eigenspace spanned by the top $d_x$ and top $d_y$ eigenvectors of $\Sigma_x$ and $\Sigma_y$.

**Extension to Nonlinear Setting.** In section 6, we mentioned that the data lying on a nonlinear manifold could lead to the apparent inconsistency between the estimator $\widehat{d}_{xy}$ being nonzero and the $p$-value produced by the hypothesis test being large. To appropriately handle the nonlinear case, a future work may consider a nonparametric extension analogous to the extension of CLVM (Severson et al., 2019) to CVAE (Abid and Zou, 2019).

## Acknowledgments and Disclosure of Funding

SH was supported by NIH grants T32ES007018 and UM1 TR004406; DL was supported by NIH grants R01 AG079291, R56 LM013784, R01 HL149683, and UM1 TR004406, R01 LM014407, P30 ES010126.

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

# A  Licensed Assets

1. BMMC data - CC BY-4.0
2. ECCITE-Seq data - Open Data Commons Open Database License (ODbL) v1.0
3. mHealth data - CC BY-4.0
4. Mouse Protein Expression data - CC BY-4.0
5. Pathogen data - Open Data Commons Open Database License (ODbL) v1.0
6. Perturb-seq data - Open Data Commons Open Database License (ODbL) v1.0
7. Small-molecule data - CC BY-4.0
8. MNIST data - CC BY-SA 3.0 license
9. contrastiveVI Github repository - BSD 3-Clause License
10. contrastive Github repository - MIT License
11. pcpca Github repository - MIT License
12. python scikit dimension package - BSD 3-Clause license
13. phthon anndata package - BSD 3-Clause License
14. python scanpy package - BSD 3-Clause License
15. python requests package - Apache-2.0 license
16. python pillow package - open source HPND License

Every contributor of new assets to this paper is an author. The authors have all mutually agreed to use the CC BY-4.0 license, as described in the README file of the Github in appendix B.

# B  Data and availability

All code and access to data pre-processing, as well as the original data sources, are available on Github at the following link.

https://github.com/myueen/contrastive-dimension-estimation

In table 3, we report the approximate time and memory requirements for each experiment. All experiments were run on a Linux-based virtual computer with 6500 conventional compute cores delivering 13,000 threads. We used only 1 core.

Table 3: Approximate Time and Memory Requirements per Experiment

| Experiment | Time | Memory |
|---|---|---|
| Simulation 1 | 5-15 minutes | 4 GB |
| Simulation 2 | 5-15 minutes | 4 GB |
| Simulation 3 | 1 hour | 4 GB |
| Simulation 4 | 1 hour | 4 GB |
| Corrupted MNIST | 1 hour | 4 GB |
| Mouse Protein | 5-15 minutes | 4 GB |
| mHealth | 1-5 minutes | 4 GB |
| BMMC | 1 hour | 160 GB |
| Small Molecule | 1 hour | 4 GB |
| ECCITE-Seq | 2 hours | 4 GB |
| Pathogen Data | 1 hour | 4 GB |
| Perturb-Seq | 2 hours | 160 GB |
| CelebA: Glasses | 6 hours | 190 GB |
| CelebA: Hat | 6 hours | 190 GB |

# C  Proofs

## C.1  Proof to lemma 1

**Lemma 1.** $d_{xy} = \dim(\mathrm{Proj}_{V_y^\perp} V_x) = \#\{i : \theta_i(V_x, V_y) > 0\} + \max(d_x - d_y, 0)$.

*Proof.* Fix $V_x, V_y \subset \mathbb{R}^p$, and let $\dim(V_x) = d_x, \dim(V_y) = d_y$. Let $\dim(V_x \cap V_y) = k$.

Let $u_1, \ldots, u_k$ be an orthonormal basis for $V_x \cap V_y$. It follows that $u_1, \ldots, u_k \in V_x \cap V_y$ are the first $k$ principal vectors for both $V_x$ and $V_y$, corresponding to principal angles $\theta_1 = \cdots = \theta_k = 0$. Furthermore, for every $i = 1, \ldots, k$, we know that $\mathrm{Proj}_{V_y^\perp} u_i = 0$, because $u_i \in V_y$.

Case 1: Assume $d_x \leq d_y$. In this case, there are $d_x - k$ nonzero principal angles corresponding to principal vectors we denote by $u_{k+1}, \ldots, u_{d_x} \in V_x$ and $v_{k+1}, \ldots, v_{d_x} \in V_y$. For each $i = k+1, \ldots, d_x$, we know that $\mathrm{Proj}_{V_y^\perp} u_i \neq 0$, because $u_i \notin V_y$. In fact, because the $u_i$ are orthogonal, $\mathrm{Proj}_{V_y^\perp} u_{k+1}, \ldots, \mathrm{Proj}_{V_y^\perp} u_{d_x}$ are a basis for $\mathrm{Proj}_{V_y^\perp} V_x$. Therefore, $\dim(\mathrm{Proj}_{V_y^\perp} V_x) = \#\{i : \theta_i(V_x, V_y) > 0\}$.

Case 2: Assume $d_x > d_y$. In this case, there are $d_y - k$ nonzero principal angles. Similar to the above argument, we can extend the basis for $V_x$ starting with the principal vectors $u_{k+1}, \ldots, u_{d_y} \in V_x$ corresponding to the principal angles. However, because there are only $d_y$ principal angles, we denote by $u_{d_y+1}, \ldots, u_{d_x}$ the remaining vectors necessary to specify an orthogonal basis for $V_x$. Because the principal angle maximizes $u_i^\top v_i$, the remaining vectors $u_{d_y+1}, \ldots, u_{d_x}$ are all orthogonal to $V_y$. Therefore, $\mathrm{Proj}_{V_y^\perp} u_{k+1}, \ldots, \mathrm{Proj}_{V_y^\perp} u_{d_y}, u_{d_y+1}, \ldots, u_{d_x}$, is a basis for $\mathrm{Proj}_{V_y^\perp} V_x$. It follows that $\dim(\mathrm{Proj}_{V_y^\perp} V_x) = \#\{i : \theta_i(V_x, V_y) > 0\} + d_x - d_y$, as desired.

$\square$

## C.2  Proof to theorem 1

**Theorem 1.** *Assume that the second moments $\Sigma_x$ and $\Sigma_y$ are finite for both groups, and that the top $d_x + 1$ eigenvalues of $\Sigma_x$ and top $d_y + 1$ eigenvalues of $\Sigma_y$ are distinct. Then, our proposed estimator $\widehat{d}_{xy}$ is consistent: $\widehat{d}_{xy} \xrightarrow[n_y \to \infty]{n_x \to \infty} d_{xy}$.*

*Proof.* We know by Anderson (1963) that the sample matrix of eigenvectors $\widehat{U}_x$ converges to the true matrix of eigenvectors $U_x$ in probability as $n_x \to \infty$, and similarly, $\widehat{U}_y \to U_y$ as $n_y \to \infty$. Let $m = \min(d_x, d_y)$, and consider the following functions.

$$
\begin{aligned}
f_1 &: \mathbb{R}^{p \times d_x} \times \mathbb{R}^{p \times d_y} \to \mathbb{R}^{d_x \times d_y} \\
&\quad (U_x, U_y) \mapsto U_x^\top U_y \\
f_2 &: \mathbb{R}^{d_x \times d_y} \to \mathbb{R}^m \\
&\quad A \mapsto (\lambda_1, \ldots, \lambda_m)^\top \text{ where } \lambda_j \text{ is the } j\text{th singular value of } A
\end{aligned}
$$

It is clear that $f_1$ is continuous. Due to Weyl's inequality for singular values, $f_2$ is continuous with respect to the spectral norm. The contrastive dimension is defined to be $d_{xy} = \#\{\lambda_j < 1\}$, while its estimator is defined by $\widehat{d}_{xy} = \#\{\widehat{\lambda}_j < 1 - \epsilon\}$. Therefore, by the continuous mapping theorem, it holds that $\widehat{\lambda} \to \lambda$ in probability. Taking $\epsilon \to 0$ gives $\widehat{d}_{xy} \to d_{xy}$, as desired. $\square$

## C.3  Proof to theorem 2

**Theorem 2.** *In addition to all assumptions in theorem 1, we assume $x$ and $y$ in model 1 are sub-Gaussian. More precisely, assume that there exist constants $K_1, K_2 \geq 1$ such that*

$$
\|\langle x, v \rangle\|_{\psi_2} \leq K_1 \mathbb{E}\left[\langle x, v \rangle^2\right] \text{ and } \|\langle y, v \rangle\|_{\psi_2} \leq K_2 \mathbb{E}\left[\langle y, v \rangle^2\right] \text{ for any } v \in \mathbb{R}^p
$$

where $\|\cdot\|_{\psi_2}$ is defined as in Vershynin (2018). Then, for any $u > 0$, with probability at least $1 - 2e^{-u}$, we have

$$\max_{j=1,\ldots,m} \left| \widehat{\lambda}_j - \lambda_j \right| \leq C \sum_{k \in \{x,y\}} \sqrt{d_k} \delta_k^{-1} K_k^2 \left( \sqrt{\frac{p+u}{n_k}} + \frac{p+u}{n_k} \right) \|\Sigma_k\| = O(n_x^{-1/2} + n_y^{-1/2})$$

(6)

where $C$ is an absolute, positive constant, $\delta$ is the minimum eigengap among the top $d+1$ eigenvalues of $\Sigma$, and $\|\cdot\|$ denotes spectral norm.

*Proof.* Let $W = U^\top V$, where $U$ is the matrix of the top $d_x$ eigenvectors of $\Sigma_x$ and $V$ is the matrix of the top $d_y$ eigenvectors of $\Sigma_y$. We use hats to represent sample estimates of these matrices and let $\widehat{W} = \widehat{U}^\top \widehat{V}$. We would like a finite-sample error upper bound on $\max_{j=1,\ldots,m} \left| \sigma_j \left( \widehat{W} \right) - \sigma_j \left( W \right) \right|$.

$$\max_{j=1,\ldots,m} \left| \sigma_j \left( \widehat{W} \right) - \sigma_j \left( W \right) \right| \leq \left\| \widehat{W} - W \right\| \quad \text{by Weyl's inequality for } \sigma\text{-values}$$

$$= \left\| \widehat{U}^\top \widehat{V} - \widehat{U}^\top V + \widehat{U}^\top V - U^\top V \right\|$$

$$\leq \left\| \widehat{U}^\top \widehat{V} - \widehat{U}^\top V \right\| + \left\| \widehat{U}^\top V - U^\top V \right\|$$

$$= \left\| \widehat{U} \right\| \cdot \left\| \widehat{V} - V \right\| + \left\| \widehat{U} - U \right\| \cdot \|V\|$$

$$= \left\| \widehat{V} - V \right\| + \left\| \widehat{U} - U \right\|$$

$$\leq \left\| \widehat{V} - V \right\|_F + \left\| \widehat{U} - U \right\|_F$$

$$\leq 2^{3/2} \left( \sqrt{d_x} \frac{\left\| \widehat{\Sigma}_{n_x} - \Sigma_x \right\|}{\delta_x} + \sqrt{d_y} \frac{\left\| \widehat{\Sigma}_{n_y} - \Sigma_y \right\|}{\delta_y} \right)$$

The final inequality holds due to the Davis-Kahan theorem, as presented in Vershynin (2018). Note that here $\delta_x$ is the smallest eigengap of $\Sigma_x$, and $\delta_y$ is the smallest eigengap of $\Sigma_y$.

For $k \in \{x, y\}$, with probability at least $1 - 2e^{-u}$, we have that

$$\left\| \widehat{\Sigma}_{n_x} - \Sigma_x \right\| \leq C K_k^2 \left( \sqrt{\frac{p+u}{n_k}} + \frac{p+u}{n_k} \right) \|\Sigma_k\|$$

where $C$ is an absolute, positive constant, and $K_x = \|x\|_{\psi_2}, K_y = \|y\|_{\psi_2}$ are the sub-Gaussian constants, as in Vershynin (2018). Substituting these upper bounds on $\left\| \widehat{\Sigma}_{n_k} - \Sigma_k \right\|$ yield the bound claimed. $\qquad \square$

## D   Additional experimental details

The code for all experiments is made available at the anonymous Github link provided in appendix B. To allow for reproducibility, we set a seed for every experiment in which we perform the bootstrap test. Sometimes we set the seed to be 1; other times we set the seed to be 42. There was no particular reason behind the choice for any seed. In every experiment requiring intrinsic dimension estimation, we used the method of moments estimator from the sci-kit dimension package. Other estimators are possible and may give slightly different results, but we chose this estimator because we found it to give the most stable and reasonable results. In our experience, we found some of the other estimators to give estimates as low as 0 and as high as the number of features, of which neither estimate is sensible in the contexts we considered.

### D.1   Simulations

For simulations 1 and 2, $S_x$ and $S_y$ were randomly chosen to be orthogonal matrices in $\mathbb{R}^{p \times d_x}$ and $\mathbb{R}^{p \times d_y}$, respectively. Therefore, the covariance matrices in this simulation are given by $\Sigma_x =$

$S_x S_x^\top + \sigma_x^2 I$ and $\Sigma_y = S_y S_y^\top + \sigma_y^2 I$, respectively. Notably, these covariance matrices violate the assumption of distinct eigenvalues presented in section 4. However, the space spanned by the top $d_x$ eigenvectors of $\Sigma_x$ is still identifiable, because the $d_x$th eigenvalue is $1 + \sigma_x^2$, while the $(d_x + 1)$th eigenvalue is $\sigma_x^2$ (similarly for $\Sigma_y$). As discussed in section 7, this is the necessary identifiability condition.

We designed simulations 3 and 4 to allow for an interpretable ground truth of $d_{xy} = 1$ while mimicking real image data. Figure 2 illustrates the data-generating process for both groups in both simulations. Every image is represented by 28 by 28 matrix in the grayscale space, where 0 represents black, and 255 represents white. Each of these image matrices was converted to a vector in $\mathbb{R}^{784}$. In both simulations, the foreground group was created by drawing a white line on top; each pixel in this line was set to 255, while the background group was created by averaging the original (disk or MNIST 0) image entrywise with an image of grass from the corrupted MNIST dataset. The was no specific intent behind choosing to draw a horizontal line in the sixth row of each image; drawing a line of random length going in any single direction from any starting pixel could have worked. However, we chose to fix the starting pixel because we wanted to be sure not to introduce a second parameter. That said, simulations in which other parameters are introduced, leaving a different ground truth $d_{xy}$ are possible. For example, creating the foreground group by drawing a white rectangle of random width and height $(w, h) \in \{1, \ldots, 24\}^2$ beginning at the top-left pixel would lead to a scenario in which $d_{xy} = 2$.

## D.2 Real data experiments

The Mouse Protein, mHealth, BMMC, Small Molecule, ECCITE-Seq, Pathogen, Perturb-Seq, and CelebA experiments all required additional pre-processing. To ensure that our pre-processing aligned with previous analyses, we used the code available for pre-processing these datasets associated with the papers analyzing them. To be specific, we pre-processed the Mouse Protein and mHealth datasets using the code associated with Abid et al. (2018). We pre-processed the Small Molecule, ECCITE-Seq, Pathogen, and Perturb-Seq datasets using the code associated with Weinberger et al. (2023). We downloaded the BMMC dataset using instructions and code from Weinberger et al. (2023) and pre-processed it using the code from Li et al. (2020). Code files for replicating the pre-processing we did for each experiment are available on the anonymous Github in appendix B. To pre-process the CelebA dataset, we first used the PIL package in Python to resize the images from 178 by 218 pixels to 64 by 64 pixels. From there, we performed PCA (Anderson, 1963) on the resized images and took the top 1000 directions.

In section 6 we made reference to our results standing in agreement with results from previous analyses. In particular, we claimed that for the BMMC dataset, our observation of $\widehat{d}_{xy} \geq 4$ was consistent with the analysis in Li et al. (2020), in which a figure was published illustrating a distinction between the pre- and posttransplant groups in the BMMC data using the third and fourth PCPCA directions. We duplicated that graph in the left panel of figure 3 for transparency. We also claimed that for the mHealth dataset, our observation of $\widehat{d}_{xy} = 1$ was consistent with the analysis in Abid et al. (2018), in which a figure was published illustrating a single direction differentiating the two groups of squatting and cycling. We duplicated this illustration in the right panel of figure 3 here for convenience.

For clarity on the image datasets used in section 6, we provide sample images from the foreground and background groups of both the Corrupted MNIST and CelebA datasets. The images for the Corrupted MNIST application are in figure 4; the top row has examples of images in the foreground group, while the bottom row has examples of images in the background group. The images for the CelebA application are in figure 5; the top-left two are foreground (glasses), while the bottom-left two are background (glasses). Similarly, the top-right two are foreground (hat), while the bottom-right two are background (hat).

## D.3 Practical Implications for Downstream Tasks

To explore how the methods we presented in this paper can be used effectively in downstream data analysis tasks, we conducted further analysis with the corrupted MNIST dataset used in section 6. We performed CPCA with dimensions $d = 1, 2, 3, \ldots, 10$, then ran logistic regression with the

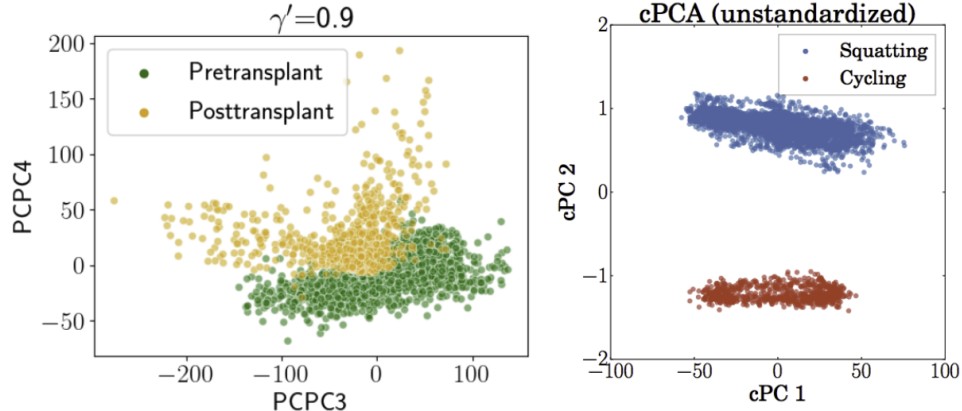

Figure 3: Left, PCPCA for BMMC Data. Right, cPCA for mHealth Data.

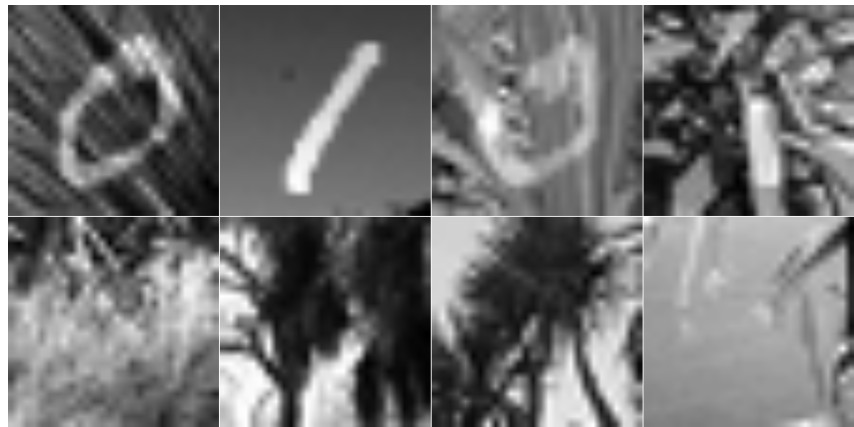

Figure 4: Corrupted MNIST Sample Images. Top row, foreground. Bottom row, background.

outcome (response) variable being the digit (0 vs. 1) imposed on the grass images. The input was the contrastive principal components (cPCs) based on CPCA with different $d$. The results are as follows:

Table 4: Summary of simulation results

| Dimension | 1 | 2 | 3 | 4 | 5 | 6 | 7 | 8 | 9 | 10 |
|-----------|-------|-------|-------|-------|-------|-------|-------|-------|-------|-------|
| Accuracy | 0.583 | 0.941 | 0.941 | 0.943 | 0.945 | 0.947 | 0.946 | 0.947 | 0.947 | 0.947 |

The accuracy increases with the number of cPCs and then plateaus at 5, where the accuracy is 0.945, after which the accuracy increases only slightly. This provides indirect evidence that if one can simply run contrastive dimension reduction methods with the dimension equal to our estimator (5 in this case). One may argue, however, that $d = 2$ is optimal, resembling an elbow point. Our principal angles (measured by singular values) align with this observation. As shown in Table 2, the smallest four singular values are 0.095, 0.315, 0.705, and 0.846. The two smallest singular values suggest that there are two more prominent contrastive dimensions, leading to satisfactory classification performance for $d = 2$. While our suggested cutoff of 0.9 gives $\hat{d}_{xy} = 5$, a cutoff of 0.7 gives $\hat{d}_{xy} = 2$. This cutoff is subjective and can be used in conjunction with classification accuracy, particularly if a response variable is available, to improve decision making.

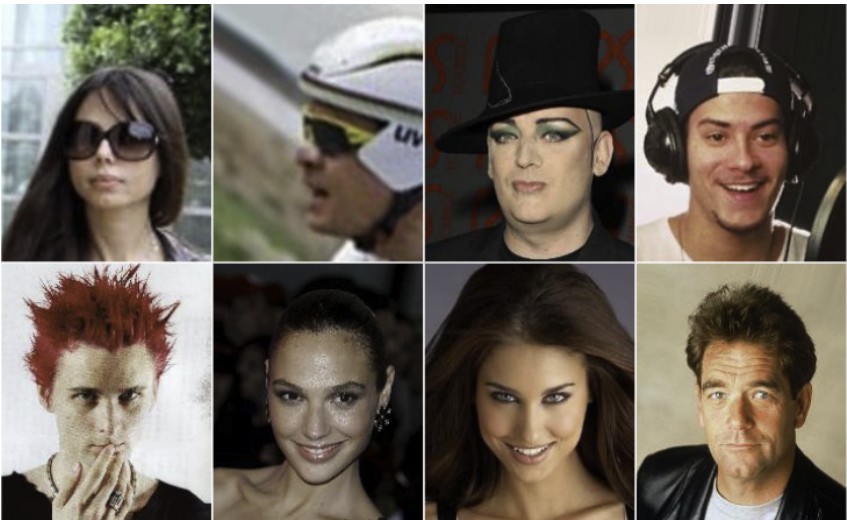

Figure 5: CelebA Sample Images. Top-left, foreground (glasses). Top-right, foreground (hat).

