# OpenReview forum: "Contrastive dimension reduction: when and how?"
_NeurIPS.cc/2024/Conference — NeurIPS 2024 poster_

### Official Review · Reviewer_Bh4k · 2024-06-28

**Soundness:** 3
**Presentation:** 3
**Contribution:** 3
**Rating:** 7
**Confidence:** 4

**Summary:**

The authors consider the problem of contrastive dimension reduction/estimation. In a nutshell, this problem involves a background dataset and a foreground dataset, and it is of interest to determine if the foreground dataset contains submanifolds/subspaces that are inherently different than those in the background. While there's past work on determining the differences of the foreground from the background, this paper asks if such a difference exists in the first place. This is posed as a hypothesis testing problem. An intuitively reasonable test is proposed and it's shown that the test is consistent. The authors also derive finite sample bounds on the error, resorting to commonly used sub-Gaussianity assumptions. They demonstrate the utility of the test through synthetic as well as real data.

**Strengths:**

The manuscript identifies a new problem in an otherwise known sub-area of dimension reduction. A solution is proposed and it is justified both asymptotically and for finite sample scenarios, using available, and accessible mathematical tools. The paper is well written, motivates the problem well, and demonstrates the solution nicely on synthetic/real world examples. I liked reading the paper and I think it would be useful for future readers.

**Weaknesses:**

I think the manuscript has minor weaknesses. The most important one of those is that the proposed test itself is not very surprising, but I don't know if that's really a weakness -- after all it's backed up theoretically, and simple is good.
Other than that, I list a few questions below, which should be easy to address by the authors.

**Questions:**

- line 81 : Do you assume a relation between $n_x$ and $n_y$, like, for instance $n_y \gg n_x$?
- line 83 : Do we center $X$, $Y$ separately? Or, in practice, do we assume $X$ and $Y$ to have the same centers?
- line 131 : Should $Z_i$ and $W_i$ be lower case? If not, why not?
- line 312 : Please state any assumptions on the dimension of $V_x$, $V_y$. If no assumptions are made, please note that.
- Defn 1 : Even a minute departure of $V_x$ from $V_y$ would result in a non-zero $d_{xy}$. But you deal with that in the sequel. It may be worth noting that.
- Defn 2 : This doesn't uniquely define $u_i$'s and $v_j$'s does it? But are $\theta_i$'s uniquely defined?
- line 174 : Again, if there are any assumptions on the ratio $n_x : n_y$ would be good to note here.
- eqn 4 : Does the expression $\text{eig}_{1:d_x}$ mean the span of the first $d_x$ eigenvectors? Please define that somewhere in the text.
- Thm 2 : It would be good to include the definition of $\| \cdot \|_{\psi_2}$, or a reference to the section where it's defined in Vershynin's book.

---

> ### Author Rebuttal · Authors · 2024-08-04
>
> Thank you for your thorough and insightful feedback on our manuscript. We appreciate the opportunity to clarify and improve our work. Below, we have summarized your questions and our responses:
>
> **Line 81 - Assumption on $n_x$ and $n_y$ and Line 174 - Ratio $n_x : n_y$**
>
> Thank you for the question and opportunity to clarify. We do not assume any specific relationship between $n_x$ and $n_y$. We have clarified this in the revised version right after we introduce the notation of $n_x$ and $n_y$.
>
> **Line 83 - Centering of $X$ and $Y$**
>
> Thank you for your question. We center $X$ and $Y$ separately in our analysis. We have clarified this in the manuscript.
>
> **Line 131 - Typographical Error**
>
> You are correct; $Z_i$ and $W_i$ should be in lowercase. We have corrected this typographical error in the revision.
>
> **Line 312 - Assumptions on Dimensions of $V_x$ and $V_y$**
>
>
> Thank you for the feedback and the opportunity to clarify. For the purpose of the contrastive dimension estimator, we do not make any assumptions on the dimension of $V_x$ and $V_y$. However, because $d_x > d_y$ implies that $d_{xy} > 0$, for the purpose of the hypothesis test, we do make the assumption that $d_x \leq d_y$: In cases where $d_x > d_y$, there is no need for a hypothesis test, and the $p$-value can be considered to be $0$.
>
> **Definition 1 - Sensitivity to Departures Between $V_x$ and $V_y$**
>
> You are absolutely correct that even a minute departure of $V_x$ from $V_y$ results in a non-zero $d_{xy}$. We have noted this in a remark right after the definition of $d_{xy}$, which motivates our study of the singular values.
>
> **Definition 2 - Uniqueness of $u_i$, $v_j$, and $\theta_i$**
>
> Indeed, $u_i$ and $v_j$ are not uniquely defined, but $\theta_i$ are uniquely defined, so principal angles are well-defined. We have clarified this right after the definition of principal angles.
>
> **Equation 4 - Definition of $\mathrm{eig}_{1:d_x}$**
>
> Yes, the expression $\mathrm{eig}_{1:d_x}$ refers to the span of the first $d_x$ eigenvectors. We have defined this notation clearly in the revision.
>
> **Theorem 2 - Definition of $\| \cdot \|_{\psi_2}$**
>
> We have included a reference to the relevant section in Vershynin's book to provide clarity in the revision.
>
> Thank you once again for your valuable feedback. We hope these clarifications and corrections will improve the clarity of our manuscript and we are available to provide further clarification or address any additional questions you may have.

---

> > ### Comment · Reviewer_Bh4k · 2024-08-12
> > **Response to author rebuttal**
> >
> > Thank you for your responses, that would help further clarify the paper. My suggestions were minor, and I already gave a high score. I'll keep it as is.

---

### Official Review · Reviewer_Casz · 2024-07-09

**Soundness:** 3
**Presentation:** 3
**Contribution:** 4
**Rating:** 7
**Confidence:** 4

**Summary:**

The paper deals with a contrastive dimension reduction (CDR) techniques which seeks to find a unique low-dimensional pattern that only exists in the foreground data compared to background data. Authors point out that despite of recent developments of related techniques such as contrastive principal component reduction (CPCA), there lacks a rigorous guideline to determine when and how those techniques should be utilized. Hence, they propose a hypothesis testing method to determine whether the foreground data actually have unique information which background data do not and an estimator of the number of CPCA PCs. The authors then investigate theoretical aspect of the estimator including the consistency and finite sample error bound. They also conduct simulation study and real data analysis to demonstrate the performance of the proposed method.

**Strengths:**

1. Contribution of introducing hypothesis testing and estimator of number of PCs to CDR literature.

2. Mathematically rigorous formulation to capture the amount of information which only exists in the foreground data.

3. Theoretical proof of the consistency and error bound of the estimator.

4. Extensive real data analysis with interpretation.

5. Well organized and well written manuscript.

**Weaknesses:**

1. Discrepancy between the p-value obtained from permutation test and the estimator happens in some cases. The authors might need to conduct additional simulation study to investigate the reason behind this discrepancy.

2. The simulation study seems to be too simple without repetitive iterations to check the stability of the proposed method. More extensive simulation is recommended. Authors need to at least provide the average of the repeatedly estimated value.


3. The proposed hypothesis testing relies on the permutation method which makes the result unstable and further theoretical investigation difficult.

**Questions:**

1. As one of the possible reasons for the discrepancy mentioned as the 1st weakness, the authors conjecture that it might be due to the nonlinearity of the underlying unique pattern. However, if there actually exists nonlinear pattern, why does the estimator succeed to capture it while hypothesis testing fails? This point needs to be elaborated.

2. From the formulation, it seems that the number of samples $n_x$ and $n_y$ should always be larger than $d_x$ and $d_y$ respectively to correctly estimate the dimension. Can you elaborate on the relationship between the sample size and intrinsic dimension with possible implications to the method?

3. Under the CLVM model which has a close relationship with CPCA, background data only has shared information implying that $d_x$ > $d_y$. However, the manuscript also concerns a case where $d_x$ < $d_y$ but $d_{xy}$ > 0 assuming the model (1). Can this be understood as suggesting a different viewpoint from CLVM to understand the CDR?

**Limitations:**

The authors well addressed the potential limitations and future research goals of the study. They included the possibility of introducing likelihood ratio testing method to overcome the weakness of the permutation test pointed out as the 3rd weakness of the manuscript.

---

> ### Author Rebuttal · Authors · 2024-08-04
>
> **Discrepancy between the p-value obtained from permutation test and the estimator happens in some cases.**
>
> Thank you for the feedback. We would like to clarify that the hypothesis test is conservative, and therefore it is possible to observe a high $p$-value even when the estimator indicates that $d_{xy} > 0$. This conservativeness is designed to reduce the likelihood of false positives, but it can also mean that some signals are not detected by the hypothesis test. We have clarified this in the revised manuscript after proposing the hypothesis test.
>
> **Nonlinear as one of the possible reasons for the discrepancy**
>
> Thank you for the question. The hypothesis test is conservative, and its reliance on resampling can lead to discrepancies when detecting nonlinear patterns. While the dimension estimator and the hypothesis test both use linear methods, they are applied differently. The estimator directly measures the dimension, which can sometimes capture variance explained by nonlinear patterns as part of the overall dimensionality. In contrast, the hypothesis test compares distributions under the assumption that $V_x \subset V_y$.
>
> To elaborate, under $H_0$ it holds that $V_x \subset V_y$. A hypothesis test that resamples the background data $y$ from both the foreground and the background (while resampling $x$ only from the foreground) creates a resampled dataset in which $V_x \subset V_y$ regardless of whether this assumption held true in the original data. Therefore, if $V_x \not\subset V_y$ in the original data, comparing the original data to the resampled data allows us to detect the desired signal.
>
> However, the hypothesis test’s resampling technique might not handle nonlinear structures in the data effectively (e.g., if the data lie on a nonlinear manifold). This can result in the test failing to detect a signal even when the estimator captures it, as the nonlinear structure can obscure the differences the test is designed to detect. We have added a remark to discuss such a discrepancy in the revised manuscript.
>
>
>
> **Additional simulations**
>
> Thank you for the feedback. We have repeated simulations 1 and 2 both 100 times and report the mean (standard deviation) of the results in the following tables, and updated our tables in the simulation section of the manuscript.
>
> | Setup  | $d_{xy}$ | $\hat{d}_{xy}$ | p-value | 4 smallest singular values |
> |--------|----------|----------------|---------|-----------------------------|
> | Sim. 1 | 0        | 0.44 (0.52)    | 0.81 (0.15) | 0.90 (0.02), 0.94 (0.01), 0.96 (0.01), 0.97 (0.01) |
> | Sim. 2 | 6        | 6 (0)          | 0.03 (0.03) | 0.06 (0.02), 0.11 (0.03), 0.16 (0.03), 0.22 (0.03) |
>
> With known ground truth, the estimator $\hat{d}_{xy}$ works perfectly for Simulation 2, while it makes some errors for Simulation 1. As we mentioned in the paper, these results indicate that a cutoff of 0.9 might be too close to 1, and a lower cutoff may be more appropriate. However, the optimal choice of cutoff could depend on the variance magnitude and the sample size.
>
> In these simulations, we used a sample size of $n_1 = n_2 = 100$ and a variance of $\sigma_x^2 = \sigma_y^2 = 0.25$. Larger sample sizes might be necessary to detect smaller angles between the subspaces $V_x$ and $V_y$. Future work could explore how varying these parameters affects the performance of the estimator and the hypothesis test, potentially leading to more precise guidelines for selecting cutoffs in different scenarios.
>
>
> **The permutation method involved in the hypothesis test**
>
> Thank you for the comments. We agree that the result from this hypothesis test can be unstable, which is why we repeat the resampling procedure 1000 times for every experiment, as discussed in Section 3.1. As mentioned in the discussion section, a likelihood-based test may be possible; however, the optimization required for it is challenging and slow. We presented the hypothesis test because it avoids the need for challenging optimization and is computationally efficient. Nevertheless, we believe that exploring a likelihood-based approach or other hypothesis tests is an interesting direction for future work.
>
>
> **The assumption of $n_x>d_x$ and $n_y>d_y$**
>
> Thank you for this question. It is correct that $n_x > d_x$ and $n_y > d_y$ are required to correctly estimate $d_{xy}$. This assumption is very reasonable, as many existing studies observed a small intrinsic dimension. For instance, Pope et.al., 2021 studied high-dimensional image data where the intrinsic dimensions typically range from 10 to 50, which is not excessively high. We have clarified this point in the revised manuscript.
>
> **Relationship between
>  $d_x$ and $d_y$. Can this be understood as suggesting a different viewpoint from CLVM to understand the CDR?**
>
> Thank you for this question. Yes, our notion of CDR can be understood from a viewpoint that differs from CLVM. CLVM assumes that $d_x \geq d_y$, ensuring unique information in the foreground group. One of our major motivations is to avoid making any assumptions about the relationship between $d_x$ and $d_y$. Note that $d_x > d_y$ implies $d_{xy} > 0$. However, even when $d_x \leq d_y$, unique information in the foreground group may still exist, which CLVM would overlook. In contrast, our method can detect such information based on our definition. We have clarified this distinction in the revision as a remark.
>
> Thank you once again for your valuable feedback. We are happy to provide further clarification or discuss any additional questions you may have.

---

> > ### Comment · Reviewer_Casz · 2024-08-12
> >
> > Thank you for the rebuttal. I'll keep my rating.

---

### Official Review · Reviewer_xiqq · 2024-07-12

**Soundness:** 2
**Presentation:** 2
**Contribution:** 2
**Rating:** 3
**Confidence:** 2

**Summary:**

The paper proposes to examine the existence and estimate the number of contrastive dimensions between the foreground and background groups. The authors provide a formal definition of contrastive dimensions and propose a hypothesis test method to examine the existence of contrastive dimensions. The authors also propose a contrastive dimension estimator to estimate the number of contrastive dimensions, and provide theoretical analysis on the consistency and finite sample error bound of the proposed estimator. Experiments are conducted on synthetic semi-synthetic and real-world datasets to evaluate the performance of the proposed hypothesis test and contrastive dimension
estimator.

**Strengths:**

1. The motivation is novel, and the proposed contrastive dimension estimator is theoretically sound.
2. The paper is well-written, and the experimental codes are made available, which make the paper easy to follow and reproduce.

**Weaknesses:**

1. The formulation of contrastive dimension seems impractical. For instance, consider the foreground group $X\in\mathbb{R}^3$ drawn from $\mathcal{N}(\mathbf{0}, diag(2, 1, \sigma_\varepsilon))$ and the background group $Y\in\mathbb{R}^3$ drawn from $\mathcal{N}(\mathbf{0}, diag(1, 1, \sigma_\varepsilon))$, where $\sigma_\varepsilon$ denotes the standard deviation of the noise terms. Both the foreground $Z_i$ and background $W_j$ occupy the same subspace, specifically the column space
$\mathcal{C}\left(\left[ {\begin{array}{ccc} 1&0&0 \\\\ 0&1&0\\\\ \end{array} } \right]^\top\right)$
. The contrastive dimension is expected to be $dim\left(\mathcal{C}\left(\left[1\quad 0\quad 0\right]^\top\right)\right)$, yet according to the definition 1, the contrastive dimension is $dim(V_{xy})=0$ .
2. In section 2, the paper claims that a significant application of the proposed methods is to determine the number of dimensions for contrastive dimension reduction. However, the experiments provide limited discussion on the effectiveness of these methods in selecting hyperparameters for downstream contrastive dimension reduction tasks. This lack of detail undermines the practical utility of the proposed
approaches.
3. The paper mentions that the hypothesis test is conservative, and the experimental results indicate that the hypothesis test does not offer significant advantages over simply detecting whether $\hat{d}_{xy}=0$. However, the paper does not justify the need for a hypothesis test when the contrastive dimension estimator appears to be effective on its own.

**Questions:**

1. Could you explain in more detail the practicality of the formulation? Can this formulation address the example given in the weaknesses section?
2. Could you conduct additional experiments to assess the effectiveness of the proposed methods in selecting hyperparameters for downstream contrastive dimension reduction tasks?
3. Could you justify the necessity of a hypothesis test when simply detecting whether $\hat{d}_{xy}=0$ seems to be effective?

**Limitations:**

Yes

---

> ### Author Rebuttal · Authors · 2024-08-05
>
> **The practicality of the formulation of contrastive dimension**
>
> Thank you for the opportunity to clarify. This example highlights the differences between our approach and methods like CPCA and CLVM.
>
> In your example, from the CLVM perspective, $\Sigma_x = SS^\top + WW^\top + \sigma_{\epsilon} \mathrm{I}$ and $\Sigma_y = SS^\top + \sigma_{\epsilon}\mathrm{I}$, where $S =  [1,0,0;0,1,0]^\top$ and $W =[1,0,0]^\top$. In this situation, both groups lie in the same 2-dimensional subspace spanned by $[1,0,0]$ and $[0,1,0]$. So, there is no low-dimensional subspace unique to the foreground; the foreground data simply has a larger variance in the direction of $[1,0,0]$ but the background also contains information in this direction. Methods like CPCA and CLVM would identify the span of $[1,0,0]$ as the space unique to the foreground, which is not valid in this case. Without further assumptions on the relationship between $S$ and $W$ in CLVM, finding a nonzero $W$ does not necessarily mean $W$ captures the linear subspace unique to the foreground—if $W = S$, it is not truly unique.
>
> In contrast, our definition gives a zero contrastive dimension in this example, which seems more reasonable. This shows our approach quantifies unique information geometrically, different from CPCA and CLVM. We are not claiming our method is more correct; rather, it is a different way to quantify unique information that might be more meaningful in certain applications. This geometric perspective may inspire new contrastive dimension reduction methods, an interesting direction for future work. We discuss this in the revised manuscript.
>
> **The effectiveness of the proposed methods in selecting hyperparameters for downstream tasks.**
>
> Measuring the effectiveness of selecting the reduced dimension hyperparameter $d$ for downstream contrastive dimension reduction tasks is indeed challenging. Contrastive dimension reduction often serves as a pre-processing step for further data analysis tasks, such as classification, clustering and visualization.
>
> Our proposed method is the first in the literature to define and estimate the contrastive dimension of a given dataset. Therefore, measuring the effectiveness of selecting the reduced dimension hyperparameter for downstream tasks is tricky. In the examples provided in the paper, we demonstrated how our methods can determine (i) whether using contrastive dimension reduction methods is appropriate, and (ii) a suggestion for the reduced (or contrastive) dimension hyperparameter in these methods.
>
> To address your request for additional experiments, we conducted further analysis using the corrupted MNIST dataset used in Section 6 in our manuscript. Here, the estimated contrastive dimension is 5. We performed CPCA with dimensions $d = 1, 2, 3, \ldots, 10$, and then ran logistic regression with the outcome (response) variable being the digit (0 vs. 1) imposed on the grass images. The input was the contrastive principal components (cPCs) based on CPCA with different $d$. The results are as follows:
>
> | Dimension (d) | 1    | 2    | 3    | 4    | 5    | 6    | 7    | 8    | 9    | 10   |
> |---------------|------|------|------|------|------|------|------|------|------|------|
> | Accuracy      | 0.58 | 0.94 | 0.94 | 0.94 | 0.95 | 0.95 | 0.95 | 0.95 | 0.95 | 0.95 |
>
> We observed that the accuracy increases with the number of cPCs and then plateaus at 5, where the accuracy is 0.95. This provides indirect evidence that if one trusts our estimator, one can simply run contrastive dimension reduction methods with the dimension equal to our estimator (5 in this case).
>
> Note that one may argue $d=2$ is optimal, resembling an elbow point. Although there is no objective way to decide, our principal angles (measured by singular values) align with this observation. As shown in Table 2 of our manuscript, the smallest four singular values are 0.095, 0.315, 0.705, and 0.846. The two smallest values, 0.095 and 0.315, suggest two more prominent contrastive dimensions, leading to satisfactory classification performance for $d=2$. If we use a cutoff of 0.7, our estimator would be $\hat{d}_{xy}=2$ as well. However, if we choose $d=5$, it aligns with our previously suggested cutoff of 0.9. The cutoff is subjective and can be used in conjunction with classification accuracy, particularly if a response variable is available, to improve decision making.
>
> Additionally, it's important to note that in the broader literature on intrinsic dimension estimation (not specifically contrastive), there is no gold standard for validating the estimated dimension. Our work similarly provides an estimator and suggests a practical approach for its application, but the validation of such estimators in an unsupervised context remains a challenging and ongoing area of research.
>
>
> **The necessity of a hypothesis test and the relationship between the test and the contrastive dimension estimator**
>
> The decision to use the hypothesis test is user-dependent. If you want to assess uncertainty, the hypothesis test provides a direct measure of whether it is worth moving forward with applying a contrastive dimension reduction method. If the test rejects the null hypothesis, it indicates that there is unique information, and estimating the contrastive dimension can guide the selection of the number of low-dimensional representations for downstream tasks.
>
> On the other hand, if users are satisfied with a point estimate of the contrastive dimension, they can rely solely on the estimator, which we have shown to be consistent and have a known error bound. We are not claiming that one must use both approaches; it depends on the user's needs and the specific application. We have clarified this in the revised manuscript, section 3.2.
>
> Thank you once again for your valuable feedback. We are happy to provide further clarification or discuss any additional questions you may have.

---

> > ### Comment · Reviewer_xiqq · 2024-08-12
> >
> > Thanks to the authors for the rebuttal. However, it does not answer my question about the practicality of their formulation, and it is not clear why the new formulation seems more reasonable (at least from the submitted manuscript or the rebuttal). I would like to keep the score unchanged.

---

> > > ### Author Response · Authors · 2024-08-12
> > >
> > > Thank you for your continued feedback. We understand the importance of demonstrating the practicality of our formulation.
> > >
> > > To clarify, our focus is not on proposing a new contrastive dimension reduction (CDR) method, but rather on determining whether or not to use a CDR method and identifying the appropriate dimension if it should be used. Existing CDR methods such as CLVM, CPCA, cVAE, can be seen as complementary, rather than competing, as they serve different purposes when analyzing case-control data.
> > >
> > >
> > > The motivation behind our definition of contrastive dimension (CD) is to ensure that the detected signal represents features truly unique to the foreground, especially in contexts where existing approaches might mistakenly identify shared features as unique. For example, consider gene expression data for diseased individuals (foreground, $x$) and healthy individuals (background, $y$). In the CLVM framework, where $\Sigma_x = SS^\top + WW^\top + \sigma^2 I_p$ and $\Sigma_y = SS^\top + \sigma^2 I_p$, the matrix $W$ is interpreted as representing genes unique to the foreground. A naive approach to defining CD might rely solely on $W$, say, the rank of $W$. However, $W$ does not necessarily contain information unique to the foreground. For example, when $S = W$, the same genes are present in both groups, and they are not truly unique to the foreground. In contrast, our method would yield a CD of zero, which is more aligned with the correct genomic interpretation—these genes are not unique to the diseased group but shared across both groups.
> > >
> > > It is important to note that CLVM does not discuss this issue and does not define any notion of contrastive dimension in their work. This potential for misinterpretation is why we defined CD in a way that specifically ensures the detected signal represents features that are truly unique to the foreground group.
> > >
> > > We do not wish to claim that this is the only way to define CD. Potential alternative definitions could be interesting and useful in different contexts. However, we believe that our definition is useful in certain cases because, by projecting onto $V_y^\perp$, we ensure that any detected signal is in a direction that is different from the space spanned by the background data, making it unique to the foreground group. As far as we know, ours is the first approach to explicitly define and estimate contrastive dimension.
> > >
> > > We believe this illustrates the utility of our formulation, particularly in applications like genomics where understanding the true uniqueness of features is critical. We are happy to provide further clarification or discuss ways to improve our work.

---

### Official Review · Reviewer_e5iP · 2024-07-13

**Soundness:** 2
**Presentation:** 3
**Contribution:** 2
**Rating:** 4
**Confidence:** 4

**Summary:**

The authors address two key challenges using two datasets, categorized as foreground and background: 1) Determine whether contrastive dimensionality methods are appropriate for application to such a pair of datasets and 2) Quantify the unique information present in the foreground data.

**Strengths:**

- The text is mostly clear including two theorems and two algorithms.
- The background section is well-written, offering a comprehensive overview of relevant literature and methodologies

**Weaknesses:**

--Hypothesis Test:
The assumptions, such as $V_x \subset V_y$ under $H_0$, lack proper justification.
The hypothesis test relies on estimating the intrinsic dimension, which is inherently challenging. The authors employ other methods for these estimates but fail to compare the accuracy of their estimates to these methods. These methods could also be used to test $d_{xy} = 0$.
The overall presentation of the hypothesis test is inadequate. The rationale for choosing this specific test and how calculating the singular values of the custom $\hat{V}_y$ improves the $p$ estimate remains unclear. The authors do not justify, explain, discuss, or comment on their choices.
--Experimental Evaluation:
The method lacks comparison to other approaches: For intrinsic dimension estimation, there's no clear way to compare against other methods due to the absence of ground truth. Alternative evaluations should be designed, such as training GANs to a priori upper-bound the intrinsic dimension of generated data by the dimension of the latent noise vector, as done in [1].
The authors don't compare their work to other "contrastive dimensionality reduction" methods. They could compare (i) expressed variance in the foreground group among different methods, (ii) expressed variance in the background group, and (iii) expressed variance common across groups. Alternatively, they could propose other ways to quantify performance and compare their method to existing literature.
The intrinsic dimension estimates are compared to those of (Pope et al., 2021), which aren't ground truth values. This makes it difficult to compare the inferred dimension to the actual one.
The practical implications of the algorithms aren't tested. How can these methods use dimensionality reduction to identify foreground sets? How can they be applied to related downstream classification tasks? How can they be used to compare different experiments?
Despite being motivated by "large scale data," the experiments are limited to small dimensions (e.g., MNIST), with no consideration of scalability crucial in today's datasets.
There's no exploration of other datasets or discussion on how intrinsic dimensions were estimated.
--Terminology: The terminology is often unclear and potentially misleading. For example, "unique information in the foreground group" refers to low-rank components, and terms like "contrastive information" and "contrastive dimension" could be misinterpreted in the context of self-supervised learning. Alternative terms like "complementary information" may be more appropriate. Even the title is misleading as "contrastive dimension reduction" is closely related to contrastive learning, a self-supervised dimensionality reduction method.
Title: The title doesn't accurately reflect the paper's content and focus.
--Missing Conclusion: The paper lacks a conclusion section.
Language Issues: The manuscript contains several typos and linguistic errors, including repeated words in the abstract (e.g., "with with").
[1] Pope, P., Zhu, C., Abdelkader, A., Goldblum, M., & Goldstein, T. The Intrinsic Dimension of Images and Its Impact on Learning. In International Conference on Learning Representations.

**Questions:**

Please see above

**Limitations:**

Please see weakness section

---

> ### Author Rebuttal · Authors · 2024-08-05
>
> **Hypothesis Test**
>
> We appreciate your feedback and the opportunity to clarify our approach.
>
> Firstly, based on Definition 1 and the notation we introduce following Model 1, the statements $d_{xy} = 0$ and $V_x \subset V_y$ are indeed equivalent. We have clarified this at the end of Section 2.
>
> We chose the method of moments estimator from the sci-kit dimension package for estimating intrinsic dimension due to its stability and reasonable results, as mentioned in Appendix D. Other estimators we considered are listed in the following table. These methods occasionally produced extreme estimates (as low as 0 or as high as $p$), which were not meaningful in our contexts.
>
> | Method | Correlation Dimension    | Dimensionality from Angle and Norm Concentration    | Expected Simplex Skewness    | Fisher Separability    | kNN (k=10)    | PCA    | Manifold-Adaptive Dimension Estimation     | Maximum Likelihood    | TwoNN   |
> |---------------|------|------|------|------|------|------|------|------|------|
> | Reference      | Campadelli, et al. (2015) | Ceruti, et al. (2012) | Johnsson, et al. (2014) | Albergante, et al. (2019) | Rozza, A., et al. (2012) | Fan, et al. (2010) | Farahmand, et al. (2007) | Levina and Bickel (2004) |  Facco, et al. (2017) |
>
> We acknowledge that estimating intrinsic dimension is inherently challenging, and this impacts the hypothesis test. Our heuristic explanation is as follows: under $H_0$, $V_x \subset V_y$. By resampling background data $y$ from both the foreground and background (while resampling $x$ only from the foreground), we create a resampled dataset where $V_x \subset V_y$ holds, regardless of its validity in the original data. Comparing the original data to the resampled data allows us to detect if $V_x \not\subset V_y$.
>
> We recognize the limitations of our hypothesis test. As discussed in section 7, a likelihood-based test may be possible, but the required optimization is challenging and slow. We presented our hypothesis test as it avoids these optimization challenges and is computationally efficient.
>
>
> **Experimental Evaluation**
>
> *Comparison with other methods*
>
> We did not compare our method to other estimates of contrastive dimension because, to our knowledge, there are no existing methods in the literature specifically for contrastive dimension estimation. We have clarified this in Section 5 and 6 in the revised manuscript.
>
> *Intrinsic dimension estimation of a single group*
>
> We acknowledge that intrinsic dimension estimation for a single dataset is an interesting problem, but it is not the focus of our paper. Our method is compatible with various intrinsic dimension estimators. For instance, using GANs to upper-bound the intrinsic dimension, as suggested, is a viable approach. Once the estimates of $d_x,d_y$ are obtained, our method can be used to find $\hat{d}_{xy}$. This flexibility allows the incorporation of different intrinsic dimension estimators, including GANs, to complement our approach. We have clarified this After Algorithm 2.
>
> *Use of contrastive DR methods*
>
> Our purpose is not to design a new contrastive DR method. Instead, we provide suggestions on whether contrastive DR methods should be used and the appropriate hyperparameter for the reduced dimension.
>
> *Practical implications and downstream tasks*
>
> Our method aids in selecting the reduced dimension, which can then be used by contrastive DR methods for various applications.
>
> *Large scale data considerations*
>
> The ten real datasets we considered have all been used in previous literature to demonstrate the usefulness of contrastive DR. While Table 2 might suggest dimensions of at most 1000, this is due to pre-processing steps like selecting the most variable genes, consistent with prior analyses of these datasets, as described in Appendix D.2. All intrinsic dimension estimates were made using the method of moments estimator from the sci-kit dimension package because it provided the most stable and reasonable results, as discussed in Appendix D.
>
> **Terminology and Title**
>
> Thank you for the helpful feedback. Our aim for the title and terminology was to reflect our focus on contrastive dimension reduction methods, such as CPCA and CLVM, and to study when it is appropriate to use them and how to improve the selection of the reduced dimension parameter.
>
> We acknowledge that terms like "contrastive information" and "contrastive dimension" might be misinterpreted within the context of self-supervised learning, where "contrastive learning" is a widely recognized concept. However, our work is rooted in the specific field of contrastive dimension reduction, which, while sharing some terminology with self-supervised learning, addresses different research questions and methodologies.
>
> Given that our work is inspired by methods like CPCA and CLVM, we have chosen terminology consistent with the established literature in this area. We believe this is the most straightforward way to communicate our contributions to researchers familiar with these methods.
>
>
> **Missing Conclusion**
>
> Thank you for your feedback. We understand the importance of having a clear conclusion section. In our current manuscript, Section 7, titled "Discussion," serves this purpose. In this section, we first summarize the contributions of our paper and then discuss possible future work. However, to make it clearer and to address your concern, we have changed the title of Section 7 to "Conclusion and Future Work" in the revised manuscript.
>
> **Typos**
>
> Thanks for catching the typos, we have fixed them in the revision.
>
> Thank you once again for your valuable feedback. We are happy to provide further clarification or discuss any additional questions you may have.

---

> > ### Comment · Reviewer_e5iP · 2024-08-11
> >
> > After considering the authors' responses and reviewing the other evaluations and rebuttals, I've decided to increase my score.

---

> > > ### Author Response · Authors · 2024-08-12
> > >
> > > Thank you for your thoughtful reconsideration and for taking the time to review our responses. We appreciate your support and are glad our clarifications were helpful. We are happy to provide any additional clarification or engage in further discussion if needed.

---

### Author Rebuttal · Authors · 2024-08-05

We thank the reviewers for their valuable feedback. This global response covers common questions and concerns. Further detailed responses are provided in the individual rebuttals.

**Hypothesis Test (e5ip, xiqq, Casz)**

Regarding the assumption of $V_x \subset V_y$ versus $d_{xy} = 0$ under $H_0$, we revised the manuscript to clarify that the statements $V_x \subset V_y$ and $d_{xy} = 0$ are equivalent.

Another question focused on the need for a hypothesis test given that an estimator is available. The decision to use the hypothesis test is user-dependent. The hypothesis test may be used as a direct measure of whether applying a contrastive DR method is worthwhile and provides a measure of uncertainty. If the test rejects the null hypothesis, it indicates that there is unique information; estimating the contrastive dimension can guide the selection of the reduced dimension parameter.


A third question involved the discrepancy possible between the results of the estimator and of the hypothesis test. We clarify that the hypothesis test is conservative, so a high p-value is possible even when the estimator indicates that $d_{xy} > 0$. This conservatism is designed to reduce the likelihood of false positives but can also lead to some signals not detected by the hypothesis test. We mentioned nonlinearity as a possible explanation for why the hypothesis test may fail to detect a signal. To elaborate, the hypothesis test’s resampling technique might not handle nonlinear structures in the data effectively (e.g., if the data lie on a nonlinear manifold). This can result in the test failing to detect a signal even when the estimator captures it, as the nonlinear structure can obscure the differences the test is designed to detect.  We discussed the possibility of using a likelihood ratio test as an alternative in the future, which could address some of these limitations. However, the optimization required for the likelihood ratio test is more complicated and computationally intensive.

**Comments Regarding Experiments (e5ip, Casz)**

One concern was the difficulty of intrinsic dimension estimation. We agree it's a challenging problem. We tried various methods but used the method of moments estimator in our analyses because it produced the most sensible results.

Another concern was the stability of the method in Simulations 1 and 2 and the need to repeat the simulation and report a measure of variation for the results. To address this concern, we repeated these simulations under the same settings 100 times and report the mean (std) of the results in the table below.

| Setup  | $d_{xy}$ | $\hat{d}_{xy}$ | p-value | 4 smallest singular values |
|--------|----------|----------------|---------|-----------------------------|
| Sim. 1 | 0 | 0.44 (0.52)  | 0.81 (0.15) | 0.90 (0.02), 0.94 (0.01), 0.96 (0.01), 0.97 (0.01) |
| Sim. 2 | 6 | 6 (0) | 0.03 (0.03) | 0.06 (0.02), 0.11 (0.03), 0.16 (0.03), 0.22 (0.03) |

With known ground truth, the estimator $\hat{d}_{xy}$ works perfectly for Simulation 2, while it makes some errors for Simulation 1. As we mentioned in the paper, these results indicate that a cutoff of 0.9 might be too close to 1, and a lower cutoff may be more appropriate. However, the optimal choice of cutoff could depend on the sample size: Larger sample sizes might be necessary to detect smaller angles between the subspaces $V_x$ and $V_y$. Future work could explore how varying these parameters affects the performance of the estimator and the hypothesis test, potentially leading to more precise guidelines for selecting cutoffs in different scenarios.

**Practical Implications for Downstream Tasks (e5ip, xiqq)**

A third issue discussed was how our methods can be used effectively in downstream data analysis tasks. To address this concern, we conducted further analysis using the corrupted MNIST dataset used in Section 6 of our manuscript. Here, the estimated contrastive dimension is 5. We performed CPCA with dimensions $d = 1, 2, 3, \ldots, 10$, and then ran logistic regression with the outcome (response) variable being the digit (0 vs. 1) imposed on the grass images. The input was the contrastive principal components (cPCs) based on CPCA with different $d$. The results are as follows:

| Dimension (d) | 1 | 2 | 3 | 4 | 5 | 6 | 7  | 8  | 9 | 10 |
|---------------|------|------|------|------|------|------|------|------|------|------|
| Accuracy | 0.58 | 0.94 | 0.94 | 0.94 | 0.95 | 0.95 | 0.95 | 0.95 | 0.95 | 0.95 |

The accuracy increases with the number of cPCs and then plateaus at 5, where the accuracy is 0.95, providing indirect evidence that one can simply run contrastive dimension reduction methods with the dimension equal to our estimator (5 in this case). One may argue, however, that $d=2$ is optimal, resembling an elbow point. Our principal angles (measured by singular values) align with this observation. As shown in Table 2 of our manuscript, the smallest four singular values are 0.095, 0.315, 0.705, and 0.846. The two smallest suggest two more prominent contrastive dimensions, leading to satisfactory classification performance for $d=2$. While our suggested cutoff of 0.9 gives $\hat{d}_{xy}=5$, a cutoff of 0.7 gives an estimate of 2. This cutoff is subjective and can be used in conjunction with classification accuracy, particularly if a response variable is available, to improve decision making.

It's important to note that in the broader literature on intrinsic dimension estimation (not specifically contrastive), there is no gold standard for validating the estimated dimension. Our work similarly provides an estimator and suggests a practical approach for its application, but the validation of such estimators in an unsupervised context remains a challenging and ongoing area of research.

We would like to thank the reviewers once again for their valuable feedback. We are open to providing further clarification or discussing any additional questions you may have.

---

### Decision · Program_Chairs · 2024-09-25

**Decision:**

Accept (poster)

**Comment:**

This paper introduces a data-driven hypothesis test to evaluate the suitability of a linear contrastive dimension reduction (CDR) model, along with an estimator for contrastive dimension. This estimate is then suggested as the reduced dimension in downstream analyses involving CDR methods. The work tackles a new problem within the well-established sub-area of dimension reduction. The authors investigate the estimator's consistency and finite sample error bounds, offering a theoretical assessment of its robustness. Validation is conducted through both synthetic examples and real-world applications. The paper is generally well-organized and well-written.

While the technical aspects appear sound, there still are some major weaknesses related to the value and practicality of the proposed CDR formulation, as well as the lack of comparison with SOTA methods to evaluate the performance of their contrastive dimension estimator. Additionally, the study is limited to linear structures in data, which the authors have acknowledged as an open problem for future work.

In the rebuttal, the authors attempted to address these issues, arguing that their work offers an alternative approach for quantifying unique information, which could be valuable in specific applications and provide complementary insights. The new formulation of contrastive dimension is motivated by ensuring that the detected signal represents unique features of the foreground. They clarified that they did not compare their method to other estimates of contrastive dimension because no existing SOTA methods specifically address contrastive dimension estimation, which appears to be correct after a non-extensive review. Nonetheless, they conducted additional numerical experiments, including one to demonstrate the effectiveness of their proposed methods in selecting the dimension hyperparameter $d$ for downstream data analysis tasks. The automatic selection of $d$ needs further discussion.

Despite the aforementioned weaknesses, which were partially addressed in the rebuttal, this paper introduces novel ideas that could contribute positively to the CDR literature. It is supported by a mathematically rigorous formulation, with codes provided to facilitate replication by the research community. Furthermore, the application of this work is particularly beneficial when dealing with datasets with a contrastive structure (e.g., in biomedical studies), where data are split into a foreground group of interest and a background group, making it potentially valuable to practitioners.

Given these considerations, the decision leans toward acceptance. However, for the camera-ready version, the authors should carefully revise the paper to position better the value of the new definition of contrastive dimension. In particular, they should further discuss the following points:

- The differences and similarities between the new task and the one addressed in traditional CDR literature.
- Whether it is possible to solve the new task, possibly with some simple post-processing techniques, using existing CDR methods.
- Potential hints on how the proposed theory could be extended to nonlinear methods.